# NOVOBENCH-100K: A LARGE-SCALE PROTEIN DATASET FOR IN SILICO EVOLUTION OF DE NOVO TADA

## ABSTRACT

We introduce NOVOBENCH-100K, a large-scale protein dataset for the in silico evolution of TadA, an enzyme critical for base editing. This dataset originates from the sequencing data collected during two rounds of our in vitro TadA evolution, encompassing 101,687 unique DNA variants with an average of 11.1 amino acid mutations. Rather than employing classes or scores as labels, our dataset consists of 77,900 ranking lists, each involving 2, 10, or 100 sequences ranked by their base editing efficiency. These rankings are generated using our SEQ2RANK, a novel algorithm that accounts for biological experiment credibility and ranking consistency. For evaluation, we provide two train-test splits, designated as in-domain ranking and out-of-domain ranking, based on a standard 7:3 random split and the actual in-vitro evolution rounds, respectively. We benchmark 80 biological language models (BLMs) across 24 papers, spanning protein, DNA, RNA, and multimodal domains. Comprehensive experiments reveal that BLMs perform well on in-domain ranking, with a detailed analysis by modality, model size, and $K$-mer. However, for out-of-domain ranking, BLMs exhibit poor performance in both linear probing and fine-tuning, resembling random guessing. This underscores the necessity for highly generalizable models to address domain shifts between experimental rounds. Finally, our wet experiments are ongoing to generate more data to expand our benchmark. In a few months, we expect to add additional rounds of in vitro evolution and include a broader variety of proteins. We will release the code, dataset, and embeddings of our evaluated 80 BLMs soon. Code and @100 dataset are provided in the supplementary.

## 1 INTRODUCTION

The remarkable progress in protein structure prediction in recent years (Jumper et al., 2021; Baek et al., 2021; Lin et al., 2023; Abramson et al., 2024; Chai Discovery team, 2024) has been largely driven by the availability of test datasets such as CASP (Moult et al., 2020), CAMEO (Haas et al., 2018), and PoseBusters (Buttenschoen et al., 2024). These datasets are essential for evaluating model performance and improving the predictive accuracy of protein structure models. While understanding protein structure offers key insights, the next critical step is designing proteins with specific functions (Chu et al., 2024; Notin et al., 2024b), an emerging and increasingly complex area of research (Ingraham et al., 2023; Watson et al., 2023; Hayes et al., 2024; DeepMind, 2024).

Gene editing has transformed biomedical science, paving the way for applications in treating genetic disorders, cancer, and viral infections. One emerging area within this domain is the base editing (Komor et al., 2016; Gaudelli et al., 2017), a more precise and safer alternative to traditional gene editing techniques (Cox et al., 2015). Base editing which converts A-T to G-C relies on tRNA-specific adenosine deaminase (TadA), an enzyme that catalyzes targeted nucleotide conversions essential for precise gene correction. Despite advances in TadA design (Ruffolo et al., 2024; Jiang et al., 2024), evaluating highly-variant TadA designs through wet lab experiments remains resource-intensive and time-consuming. This highlights the need for high-quality datasets for in silico evaluation to accelerate the functional protein design of TadA.

In this paper, we present NOVOBENCH-100K, a large-scale protein dataset for the in silico evolution of de novo TadA. It includes 101,687 unique DNA variants derived from biological sequencing data in our Phage-Assisted Non-Continuous Evolution (PANCE) experiments, as illustrated in Figure 1.

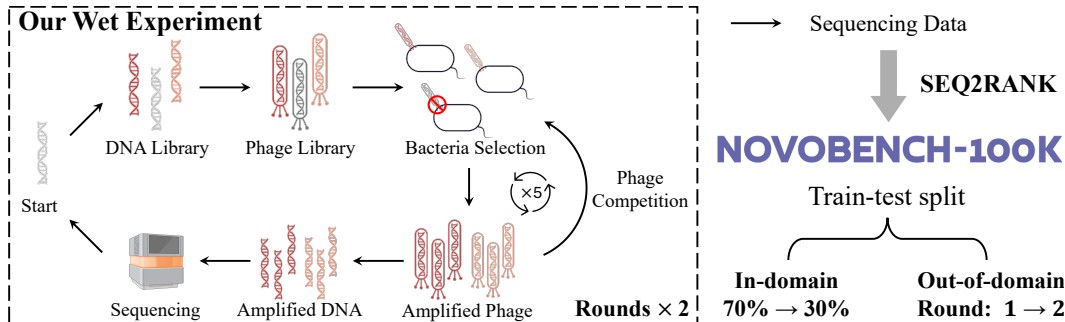

Figure 1: **We construct a large-scale protein dataset, NOVOBENCH-100K, based on our multiple rounds of wet experiments.** We employ a standardized experimental procedure of Phage-Assisted Non-Continuous Evolution (PANCE). For evaluation, two train-test splits are offered, in-domain ranking split by a standard 7:3 ratio, and out-of-domain ranking split by actual evolution rounds.

Unlike deep mutation scanning methods (Sumi et al., 2024) with limited mutations, our benchmark introduces an average of 11.1 amino acid mutations compared to the initial sequence used for evolution. Rather than taking classes or scores as labels, we propose using rankings in biological datasets for less experimental noise and better dataset extension. Our dataset consists of 77,900 ranking lists sorted by their base editing efficiency, divided into three tracks featuring ranking lists of varying lengths—2, 10, and 100—to accommodate different levels of ranking difficulty.

Our dataset is constructed using our novel algorithm, SEQ2RANK, which efficiently transforms large-scale biological sequencing data into consistent ranking lists. Firstly, this algorithm ensures experimental-level consistency by sorting sequencing data based on their credibility. Such credibility is informed by biological knowledge (*e.g.*, the decreasing influence of initial randomness over time) and experimental indicators (*e.g.*, gel electrophoresis). Moreover, we employ a directed acyclic graph to maintain strict sequence-level consistency across ranking lists. This structure provides an essential foundation for machine learning models to identify meaningful patterns within datasets.

NOVOBENCH-100K evaluates model performance through a ranking task in which models are required to rank sequences based on their editing efficiency within each sequence list. We offer two train-test splits on the same dataset, referred to as in-domain ranking and out-of-domain ranking. The in-domain ranking takes a standard 7:3 random split on all our ranking lists for each track, while out-of-domain ranking is based on actual in-vitro evolution rounds. The data distribution across different rounds of evolution may vary significantly. The latter split is significantly more challenging as it captures the real dynamics of biological evolution, partitioning the train-test sets according to actual experimental rounds. Training a model on the training set simulates the practical scenarios in protein evolution, where the outcomes of future rounds, corresponding to our test set, are unknown.

We benchmark 80 biological language models (BLMs) across 24 papers on our NOVOBENCH-100K using linear probing and fine-tuning under in-silico evolution scenarios, shown in Figure 2. A 3-layer fully connected layer is taken for the ranking task using the ListNet loss (Cao et al., 2007). Considering that DNA, RNA, and proteins function as an integrated unit, it is reasonable to transcribe or translate the original DNA sequencing data in NOVOBENCH-100K to other biological "languages." Therefore, our experiments span BLMs among multiple modalities, including proteins, DNA, RNA, and multimodalities. Specifically, we extract features of the last trunk for folding models such as Chai1 (Chai Discovery team, 2024) and RoseTTAFold-All-Atom Krishna et al. (2024).

Our results indicate that current BLMs perform well on in-domain ranking, achieving high scores on metrics such as normalized discounted cumulative gain (nDCG) and Spearman's rank correlation (SP). We conduct a comprehensive analysis to explore how modality, model size, and $K$-mer, influence the performance on NOVOBENCH-100K. However, in out-of-domain ranking, all BLMs perform poorly using linear probing and fine-tuning, with results comparable to random guessing. We attribute this to the significant domain gap between in-vitro evolution rounds, as evidenced by decreasing training loss while test metrics remain unchanged. This highlights the need for highly generalizable models to handle "out-of-domain" tasks prevalent in real-world applications, such as protein evolution.

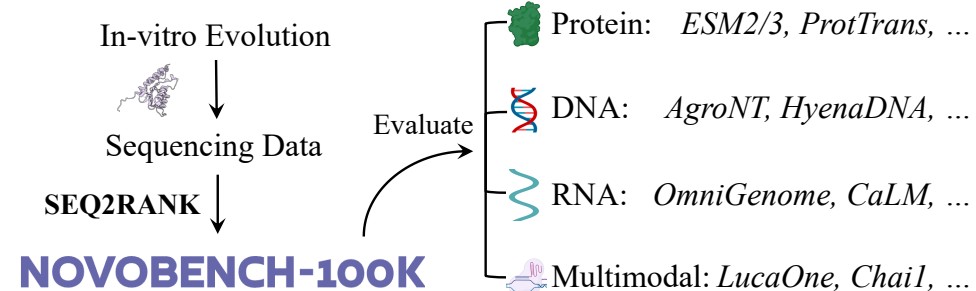

Figure 2: **NOVOBENCH-100K is constructed using our SEQ2RANK algorithm based on biological sequencing data collected from our in vitro TadA evolution.** We evaluate 80 biological language models across 24 papers, covering protein, DNA, RNA, and multimodal domains.

**Our contributions:**

- We propose NOVOBENCH-100K, a large-scale protein dataset derived from our wet lab experiments, for the in silico evolution of de novo TadA. Our dataset encompasses 101,687 unique DNA variants collected from in vitro evolution experiments.

- We propose a novel algorithm, SEQ2RANK, to transform biological sequencing data into consistent rankings. It effectively integrates the credibility of various experiments and ensures strict ranking data consistency.

- We benchmark 80 BLMs in 24 papers on our dataset. Results show that while BLMs perform well on in-domain ranking, they universally fail on out-of-domain ranking using linear probing and fine-tuning, highlighting the need for highly generalizable models.

## 2 RELATED WORK

### 2.1 BASE EDITING DATASET

Current datasets focusing on deaminase enzyme optimization in base editing are limited. Most existing datasets (Dixit et al., 2024; Yan et al., 2020; Marquart et al., 2021; Sánchez-Rivera et al., 2022; Xiang et al., 2021; Leenay et al., 2019) concentrate on refining the interactions between deaminases and ancillary elements like Cas proteins and sgRNAs. While numerous labs often publish their improved deaminase protein sequence (Richter et al., 2020; Li et al., 2020; Tu et al., 2022; Perrotta et al., 2024; Cheng et al., 2024), there are not enough experimental settings public, thus hampering the independence and identically distributed assumption in machine learning applications. Additionally, CRISPRbase (Fan et al., 2023) aggregates existing datasets from various labs and introduces significant batch effects due to diverse experimental protocols (Notin et al., 2024a), compromising the accuracy of models under real experimental conditions. We introduce NOVOBENCH-100K, designed to simulate real-world laboratory conditions more accurately and consistently, thereby enhancing the precision in deaminase evolution and leaving space for more NGS data in the future.

### 2.2 PROTEIN FUNCTION BENCHMARK

While structural benchmarks in protein research have achieved notable success (Ye et al., 2024; Moult et al., 2020; Haas et al., 2018; Buttenschoen et al., 2024), functional benchmarks are still in nascent stages. These benchmarks are primarily categorized into two groups (West-Roberts et al., 2024), biophysical properties and deep mutational scanning (DMS) data. The benchmarks for biological properties (Bairoch, 2000; Xu et al., 2022; Zhou et al., 2019; Nikam et al., 2021; Rao et al., 2019; Vander Meersche et al., 2024) include metrics like enzymatic activity, fluorescence, thermodynamics, and solubility; however, their broad focus limits their utility in precise evaluations. DMS benchmarks (Fowler & Fields, 2014; Jiang et al., 2024; Gray et al., 2018), which utilize large-scale mutagenesis and high-throughput sequencing, offer detailed insights into fitness landscapes for protein mutations. Researchers (Notin et al., 2024a; Dallago et al., 2021; Riesselman et al., 2018) also leverage diverse DMS datasets to construct the comprehensive benchmark but may introduce

Figure 3: **Our NOVOBENCH-100K dataset offers three key advantages over related benchmarks.** It is specifically collected for practical application of TadA evolution. It involves 101,687 unique deaminase variants with an average of 11.1 amino acid mutations. Our standardized wet experiments and novel algorithm SEQ2RANK ensure general and strict data consistency.

inconsistency since the way data is treated varies widely within the community (Notin et al., 2024a). In conclusion, the above benchmarks are for general purposes without specific application. In contrast, NOVOBENCH-100K is designed especially for the TadA protein evolution. Besides, it guarantees strict consistency among data with average mutation of 11.1 amino acids, as illustrated in Figure 3.

## 3 NOVOBENCH-100K

This section presents the construction and attributes of our dataset, NOVOBENCH-100K. Initially, we provide essential biological context to enhance understanding in Section 3.1. We then detail the procedures for gathering and processing raw sequencing data in Section 3.2. Additionally, we introduce our novel algorithm, SEQ2RANK, which converts biological sequencing data into rankings in Section 3.3. This algorithm ensures consistency and diversification, accounting for the varying reliability of wet lab experiments. Finally, we provide characteristics of NOVOBENCH-100K in Section 3.4, illustrating the dataset's basic statistics and comprehensive utility.

### 3.1 BIOLOGICAL BACKGROUND

Gene editing is a groundbreaking biotechnological approach that allows for precise DNA alterations, offering transformative potential in treating genetic disorders, enhancing agricultural practices, and advancing personalized medicine. A key challenge in gene editing involves precise base-level modifications, a task for which base editing is specifically designed. Unlike traditional CRISPR methods that cut DNA (Cox et al., 2015; Hilton & Gersbach, 2015), base editors modify single DNA bases without inducing double-strand breaks, providing a safer and more precise alternative (Komor et al., 2016; Gaudelli et al., 2017). Base editors are categorized into cytosine base editors, which convert C-G to T-A, and adenine base editors (ABEs), which convert A-T to G-C. ABEs utilize the tRNA-specific adenosine deaminase (TadA), an enzyme that catalyzes targeted nucleotide conversions crucial for accurate gene correction. TadA8e (Richter et al., 2020) represents an evolved form of TadA, capable of converting adenine to inosine [1], facilitating precise A-to-G edits.

Evolving TadA for enhanced editing efficiency is crucial as it directly influences the therapeutic potential and specificity of gene editing tools, potentially revolutionizing treatments for genetic diseases by ensuring more accurate and efficient genomic interventions. In this context, AI-driven models can significantly expedite the discovery of effective evolutions, optimizing the enzyme design process across extensive search spaces. Hence, we build NOVOBENCH-100K by collecting

---

[1]Inosine is interpreted as guanine in DNA.

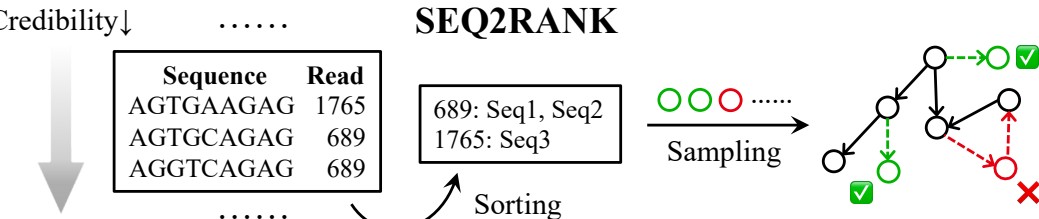

Figure 4: **We propose a novel algorithm SEQ2RANK to transform large-scale biological sequencing data into ranking lists.** It offers experiment-level consistency by prioritizing experiments based on their credibility, such as biological knowledge and experimental indicators. Additionally, it adopts a directed acyclic graph to ensure strict sequence-level consistency among ranking lists.

sequencing data in our Phage-Assisted Non-Continuous Evolution (PANCE) experiments, details of which can be found in Appendix A.1.

## 3.2 SEQUENCING DATA TACKLING

All data in NOVOBENCH-100K originate from the PANCE of a well-known TadA enzyme, TadA8e (Richter et al., 2020). In our experiments, the base editing system is engineered to link the activity of various deaminase mutants to their amplification efficiency directly: higher deaminase activity leads to faster proliferation of bacteria, thereby enriching variants with elevated activity. During these experiments, we gather raw next-generation sequencing (NGS) data of the extracted DNA from lysed bacteria, which provides an exhaustive snapshot of genetic sequences, facilitating the high-throughput analysis of DNA. This raw NGS data quantifies the prevalence of different DNA sequences, reflecting the editing efficiencies of specific TadA proteins. We process the raw NGS data using CRISPResso2 (Clement et al., 2019), which includes standard procedures such as quality filtering, adapter trimming, read merging, and alignment. Ultimately, we generate "sequence-read" pairs, where the "read" of each sequence denotes the count of sequences detected during NGS, indicative of the editing efficiency of particular TadA variants.

## 3.3 SEQ2RANK

Typically, after processing sequencing data to obtain "sequence-read" pairs, these pairs are directly used to build a regression dataset (Zhao et al., 2021; Mortazavi et al., 2008). However, using absolute read counts as labels introduces several risks. First, these values are highly susceptible to experimental noise, including measurement sensitivity, instrumental variability, operator variability, and environmental conditions. Second, it limits the dataset's scalability and complicates the integration of data across different studies due to batch effects in biological experiments.

Instead, we propose using rankings as a more robust and scalable label format, similar to those used in recommendation systems (Qin et al., 2010). Our data unit consists of sequences ordered by read number, indicating editing efficiency in base editing as higher-ranked sequences demonstrate greater protein activity, as discussed in Section 3.2. This approach shifts the task to predicting the correct sequence rankings rather than individual efficiency scores.

However, constructing a ranking dataset from NGS data presents new challenges. Firstly, it is challenging to handle the vast amount of NGS data with the sensitivity issues inherent to biological sequencing. Additionally, conflicting rankings frequently arise across different experimental rounds due to variations in biological procedures, with these experiment-level conflicts often depending on the credibility of the evolution rounds. Finally, as a form of partial ordering, ranking can introduce potential conflicts among sequences across ranking lists due to transitivity. These hidden sequence-level conflicts within datasets can adversely affect the effectiveness of machine learning models.

To address these issues, we propose SEQ2RANK to transform NGS lists to ranking lists, as illustrated in Figure 4. We employ a greedy sampling strategy to manage the tens of thousands of sequences within each NGS list. During the sampling, we construct a "read-sequence" dictionary and ensure

one unique read key can be sampled only once within each ranking list. Such a "strict" partial order can relieve measurement sensitivity issues.

Furthermore, our algorithm mitigates experiment-level conflicts by prioritizing experiments based on their credibility, which can be informed by biological knowledge. For example, later experimental rounds are considered more reliable, as the effects of initial randomness decrease over time. Credibility can also be assessed through validation techniques such as gel electrophoresis, qPCR, and Sanger sequencing, providing references for the success and stability of a specific biological experiment. Leveraging such credibility, we sort sequencing data by credibility, reducing round-level conflicts.

Finally, we adopt a directed acyclic graph (DAG) to ensure strict sequence-level consistency in the rankings. In the DAG, we take each sequence as a node and the partial order relationship as a directed edge. A new sequence to sample is considered "safe" when it will not introduce a circle in our DAG. When generating a ranking list, we will iteratively sample unselected keys, *i.e.*, the read numbers, of the "read-sequence" dictionary. For each read number, all corresponding sequences will be checked until a "safe" sequence is found.

SEQ2RANK effectively integrates prior biological knowledge and guarantees data consistency, making it suitable for handling large-scale biological sequencing data. This integration significantly enhances the quality of genomic interpretations, leading to more reliable and actionable biological insights. The detailed pseudocode algorithm can be found in Appendix A.2.

### 3.4 CHARACTERISTICS

Utilizing SEQ2RANK, we have effectively constructed NOVOBENCH-100K, which includes 101,687 unique TadA variants derived from NGS data in base editing evolution experiments. The in-vitro evolution starts with TadA8e where the system is designed to correlate the activity of TadA mutants directly with their amplification efficiency—higher deaminase activity results in quicker bacteria proliferation, enriching for more active variants. Unlike deep mutation scanning approaches (Sumi et al., 2024), NOVOBENCH-100K incorporates an average of 11.1 amino acid changes compared with the original sequences used for evolution.

NOVOBENCH-100K assesses model performance through a ranking task that challenges models to order sequences based on their editing efficiency. Considering TadA functions across the biological spectrum from DNA to RNA, and then to protein, it is beneficial to include corresponding protein and RNA sequences for a comprehensive evaluation of protein and RNA models. This "sequence-to-function" approach is ideally suited for biological language models (BLMs) that are pre-trained on a variety of biological languages, including protein, DNA, and RNA sequences, enhancing their ability to generalize across different biological tasks.

Distinct from traditional protein function datasets, NOVOBENCH-100K emphasizes ranking sequences based on relative performance instead of absolute metrics, which are frequently distorted by batch effects and experimental inconsistencies. This ranking approach mitigates the impact of variable experimental conditions and ensures that the dataset more accurately represents genuine biological phenomena. Focusing on relative performance also reduces the influence of experimental noise, such as variations in instrument calibration, reagent quality, and operator handling. Moreover, this strategy boosts the dataset's scalability across different experimental setups and enhances its adaptability for integration with data from various sources.

### 3.5 TRAIN-TEST SPLIT

NOVOBENCH-100K supports three distinct evaluation tracks with ranking lists of varying lengths—2, 10, and 100—to suit different analytical depths required by various BLMs. Each track offers two train-test splits on the same dataset, referred to as in-domain ranking and out-of-domain ranking. The in-domain ranking takes a standard 7:3 random split on all sequence data, while out-of-domain ranking is based on actual in-vitro evolution rounds. The latter split, shown in Appendix Table 7, is significantly more challenging as it captures the real dynamics of biological evolution, partitioning the train-test sets according to actual experimental rounds. Training a model with this dataset mirrors real-world scenarios of protein evolution, where the outcomes of subsequent experimental rounds, akin to the test set in NOVOBENCH-100K, are unknown.

# 4 EXPERIMENTS

First, we outline our experimental settings in Section 4.1, detailing the biological language models (BLMs) and evaluation metrics used. Next, we present the performance of BLMs on in-domain ranking, analyzing the impact of modality, model size, and $K$-mer in Section 4.2. Finally, we demonstrate the limitations of BLMs on out-of-domain ranking, exploring their performance under both linear probing and fine-tuning in Section 4.3.

## 4.1 SETTINGS

### 4.1.1 BIOLOGY LANGUAGE MODEL

The evaluation is based on protein, DNA, and RNA modalities, since these 3 forms play important roles in the natural transcription and translation process, and all contain important information. In NOVOBENCH-100K, DNA sequences obtained from biological sequencing data are translated into RNA and protein sequences according to biological principles. These transformed sequences are inputs for the corresponding biological language models (BLMs).

As for protein modality, we test the ESM2 (Lin et al., 2023), ESM3 (Hayes et al., 2024), Prot-Trans (Elnaggar et al., 2021), SaProt (Su et al., 2023), and RFAA (Krishna et al., 2024). Our DNA modality evaluation involves the EVO (Nguyen et al., 2024a), NucleotideTansformer (NT) (Dalla-Torre et al., 2023), AgroNT (Mendoza-Revilla et al., 2024), GenSLMs (Zvyagin et al., 2023), HyenaDNA (Nguyen et al., 2024b), DNABERT-1 (Ji et al., 2021), DNABERT-2 (Zhou et al., 2023), and DNABERT-S (Zhou et al., 2024). For RNA modality, NOVOBENCH-100K tests the RNA-FM (Chen et al., 2022), SpliceBERT (Chen et al., 2023), 3UTRBERT (Yang et al., 2023), OmniGenome (Yang & Li, 2024), CaLM (Outeiral & Deane, 2024), ERNIE-RNA (Yin et al., 2024), RNAErnie (Wang et al., 2024), RNA-MSM (Zhang et al., 2024), and RiNALMo (Penić et al., 2024). We also include LucaOne (He et al., 2024) and Chai1 (Chai Discovery team, 2024) as representatives of multimodal BLMs, reflecting the popular concept of multimodality in the foundation models domain.

Overall, we test 80 models across 24 papers [2]. For linear probing, we use multimodal BLMs such as LucaOne and Chai1 to tackle three modalities of input sequence input independently, referred to as three BLMs for convenience. We extract features of the last trunk for folding models such as Chai1 and RoseTTAFold-All-Atom. Owing to space limitations, we only report one model for each paper in Table 1. The complete evaluation of 80 models can be found in the Appendix Tables 3 to 5.

### 4.1.2 RANKING EVALUATION METRICS

We offer three tracks, @2, @10, and @100 for different ranking lists with corresponding lengths. Two train-test splits, designated as in-domain ranking and out-of-domain ranking, are provided based on a standard 7:3 random split and the actual in-vitro evolution rounds, respectively. We take a 3-layer fully connected network with a hidden size of 128 as the head module, using a cross-entropy ListNet loss (Cao et al., 2007). We adopt linear probing and fine-tuning to evaluate the performance of various BLMs without introducing complex structures in the head module.

We adopt three common ranking evaluation metrics to assess the effectiveness of the predicted rankings within a population of size $x$, normalized discounted cumulative gain (nDCG@$x$) (Järvelin & Kekäläinen, 2000), mean Reciprocal Rank (mRR@$x$) (Wu et al., 2011), and Spearman's Rank Correlation (SP@$x$) (Sedgwick, 2014). The nDCG measures the accuracy of ranking results, with greater emphasis placed on higher-ranked items. The mRR focuses exclusively on the accuracy of predictions for the top-ranked sample, aligning closely with objectives in protein evolution. SP evaluates the predicted rankings' overall distribution. Details can be found in Appendix A.3.

## 4.2 IN-DOMAIN RANKING

For the in-domain ranking task, we primarily take the linear probing with a batch size of 64, freezing the parameters of BLMs, and using the output embeddings to train head modules. The

---

[2]There are some other BLMs that we do not include, such as Atom-1 (Boyd et al., 2023), UNI-RNA (Wang et al., 2023), and RFamGen (Sumi et al., 2024), since their codebases or model weights have not been released.

Table 1: **Diverse BLMs are evaluated on three tracks using linear probing under in-domain ranking.** The top, middle, and bottom groups are protein, DNA, and RNA BLMs. We report the result of one model from each model family of 24 papers. We take using one-hot vectors of sequences as baselines. * indicates that a smaller batch size is employed due to the large size of the embeddings.

| Model | @2 | | | @10 | | | @100 | | |
|---|---|---|---|---|---|---|---|---|---|
| | nDCG↑ | mRR↑ | SP↑ | nDCG↑ | mRR↑ | SP↑ | nDCG↑ | mRR↑ | SP↑ |
| One-hot | 0.826 | 0.764 | 0.058 | 0.820 | 0.322 | 0.079 | 0.854 | 0.057 | -0.009 |
| Chai1 | 0.847 | 0.792 | 0.169 | 0.857 | 0.322 | 0.194 | 0.900 | 0.095 | 0.138 |
| ESM2 | 0.831 | 0.771 | 0.082 | 0.844 | 0.322 | 0.175 | 0.907 | 0.050 | 0.252 |
| ESM3 | 0.840 | 0.783 | 0.133 | 0.860 | 0.335 | 0.214 | 0.892 | 0.100 | 0.103 |
| RFAA | 0.838 | 0.780 | 0.120 | 0.858 | 0.323 | 0.205 | 0.890 | 0.050 | 0.158 |
| SaProt | 0.831 | 0.771 | 0.083 | 0.839 | 0.322 | 0.144 | 0.864 | 0.042 | 0.085 |
| LucaOne | 0.830 | 0.770 | 0.078 | 0.839 | 0.313 | 0.146 | 0.901 | 0.029 | 0.218 |
| ProtTrans | 0.831 | 0.771 | 0.085 | 0.844 | 0.325 | 0.172 | 0.886 | 0.038 | 0.185 |
| One-hot | 0.819 | 0.754 | 0.017 | 0.822 | 0.281 | 0.072 | 0.854 | 0.052 | 0.027 |
| NT | 0.836 | 0.777 | 0.109 | 0.845 | 0.307 | 0.180 | 0.884 | 0.030 | 0.182 |
| EVO* | 0.830 | 0.770 | 0.080 | 0.850 | 0.317 | 0.190 | 0.895 | 0.007 | 0.153 |
| Chai1 | 0.848 | 0.794 | 0.175 | 0.868 | 0.322 | 0.224 | 0.901 | 0.086 | 0.222 |
| AgroNT | 0.831 | 0.772 | 0.086 | 0.839 | 0.304 | 0.156 | 0.868 | 0.096 | 0.123 |
| GenSLM* | 0.836 | 0.777 | 0.109 | 0.857 | 0.327 | 0.204 | 0.904 | 0.043 | 0.233 |
| LucaOne* | 0.835 | 0.776 | 0.106 | 0.843 | 0.303 | 0.165 | 0.888 | 0.037 | 0.165 |
| HyenaDNA | 0.831 | 0.771 | 0.085 | 0.848 | 0.314 | 0.178 | 0.883 | 0.029 | 0.132 |
| DNABERT-2 | 0.816 | 0.750 | 0.001 | 0.814 | 0.295 | 0.038 | 0.861 | 0.026 | 0.018 |
| DNABERT-S | 0.817 | 0.752 | 0.007 | 0.812 | 0.301 | 0.028 | 0.853 | 0.040 | 0.017 |
| DNABERT-1 | 0.835 | 0.776 | 0.105 | 0.845 | 0.299 | 0.163 | 0.893 | 0.075 | 0.236 |
| One-hot | 0.819 | 0.754 | 0.017 | 0.822 | 0.281 | 0.072 | 0.854 | 0.052 | 0.027 |
| Chai1 | 0.845 | 0.790 | 0.161 | 0.867 | 0.316 | 0.225 | 0.897 | 0.124 | 0.217 |
| CaLM | 0.834 | 0.775 | 0.099 | 0.847 | 0.309 | 0.178 | 0.882 | 0.062 | 0.146 |
| RNA-FM | 0.830 | 0.770 | 0.079 | 0.846 | 0.315 | 0.187 | 0.880 | 0.026 | 0.117 |
| RiNALMo* | 0.843 | 0.787 | 0.148 | 0.870 | 0.326 | 0.235 | 0.904 | 0.049 | 0.198 |
| RNAErnie* | 0.837 | 0.780 | 0.119 | 0.867 | 0.326 | 0.230 | 0.906 | 0.056 | 0.237 |
| RNA-MSM | 0.832 | 0.773 | 0.090 | 0.850 | 0.317 | 0.197 | 0.902 | 0.045 | 0.253 |
| SpliceBERT | 0.833 | 0.774 | 0.095 | 0.844 | 0.308 | 0.167 | 0.890 | 0.051 | 0.203 |
| 3UTRBERT* | 0.840 | 0.784 | 0.135 | 0.870 | 0.324 | 0.244 | 0.908 | 0.044 | 0.256 |
| ERNIE-RNA* | 0.836 | 0.778 | 0.113 | 0.860 | 0.327 | 0.230 | 0.909 | 0.041 | 0.213 |
| OmniGenome* | 0.838 | 0.781 | 0.122 | 0.868 | 0.320 | 0.239 | 0.910 | 0.059 | 0.207 |

fine-tuning experiments for selected models can be found in Appendix Table 6. We have examined the hyperparameters such as batch size, number of training epochs, and head model architecture [3]. Given the influence of different embedding lengths on the learning rate, we specify 3 learning rates for each experiment, 1e-5, 1e-4, and 1e-3, and choose the optimal result as its reported result. We report the performance of each BLM family on in-domain ranking task in Table 1 for ranking lists with different lengths of 2, 10, and 100. We use one-hot vectors of the sequences as the baseline to compare with embeddings of BLMs.

Compared to training classification heads directly using sequence one-hot vectors, using embeddings extracted from pre-trained BLMs significantly enhances the test performance. This demonstrates that BLMs are well-suited for in-domain ranking tasks on NOVOBENCH-100K, aligning with experiences in the language model field. The complete evaluation of 80 models can be found in the Appendix Tables 3 to 5. We have also fine-tuned the BLMs (as shown in Appendix Table 6), which further improves performance. This aligns well with a general understanding of language models, while it is not the main focus of this paper.

---

[3]It is worth noting that a smaller batch size will be employed due to the large size of some embeddings. We have reported the hyperparameters examination results in Appendix Table 2, and we have also included the analysis of data precision in that section.

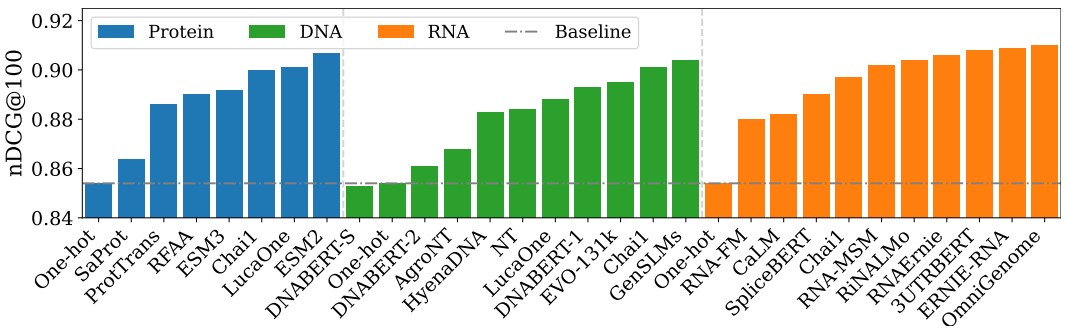

Figure 5: **BLMs using different modalities perform comparably using linear probing on the @100 track.** The performance gap between the 3 modalities of BLMs is not obvious, which means the knowledge of DNA and RNA BLMs is also important in the protein evolution task.

### 4.2.1 MODALITY

NOVOBENCH-100K primarily challenges BLMs in predicting TadA function, making it a protein evolution task. Interestingly, DNA and RNA BLMs demonstrate performance comparable to protein BLMs on nDCG@100, as shown in Figure 5. It demonstrates that the nucleotide BLMs also gain knowledge about protein functionality on DNA or RNA sequences. Since proteins, DNA, and RNA fundamentally form an integrated within organisms and each plays a crucial role in protein expression, models across all three modalities significantly outperform those trained on one-hot vectors of the sequences.

Furthermore, it is insightful to explore the performance differences across modalities within a single multimodal BLM, given their emergence as powerful tools in recent years. We evaluate LucaOne (He et al., 2024) and Chai1 (Chai Discovery team, 2024) on NOVOBENCH-100K. LucaOne achieves nDCG@100 scores of 0.901 for protein and 0.888 for nucleotide [4], whereas Chai1 attains scores of 0.900 and 0.897, respectively. These results indicate that Chai1 achieves better modality unification, as its performance across modalities is more consistent, while LucaOne shows a notable advantage in protein performance over nucleotide.

### 4.2.2 MODEL SIZE

The prevalent view that larger models yield deeper understanding, termed the "scaling law" (Kaplan et al., 2020), has been widely accepted. However, whether such phenomena exist for biological language models in protein, DNA, and RNA modalities remains unknown. Therefore, we investigate whether BLMs exhibit similar characteristics on NOVOBENCH-100K. Most BLM model families in Figure 6 demonstrate the scaling law. More results can be found in the Appendix Tables 3 to 5.

### 4.2.3 K-MER

*K*-mer in BLMs sequence of *k* consecutive nucleotides used to capture local sequence patterns and the context in biological modeling analysis. 3UTRBERT is an RNA BLM model family composed of different *k*-mer models. Considering the test nDCG@10 in in-domain ranking, the results for 6-mer, 5-mer, 4-mer, and 3-mer are respectively 0.870, 0.860, 0.869, and 0.870. We observe that the results for 3-mer and 6-mer are higher than those for 4-mer and 5-mer. In biological terms, a protein is encoded by three nucleotides, demonstrating that NOVOBENCH-100K aligns well with the actual biological *k*-mer patterns. It also indicates that RNA BLMs are significant in protein-related tasks, provided that an appropriate *k*-mer is selected.

### 4.3 OUT-OF-DOMAIN RANKING

The in-domain ranking indeed conforms to the rules in machine learning, but such approaches may not align with practical scenarios in protein evolution. In the process of protein evolution, results

---

[4]LucaOne treats DNA and RNA equivalently by mapping "T" and "U" to the same token.

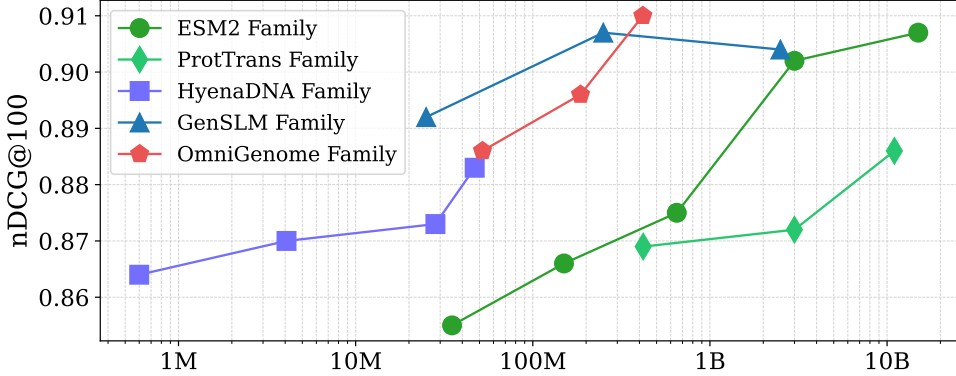

Figure 6: **The scaling law behavior is demonstrated for selected BLM families among three modalities.** We select BLM families across three modalities, protein, DNA, and RNA. The x-axis represents the parameter number and the y-axis reflects the nDCG@100 score.

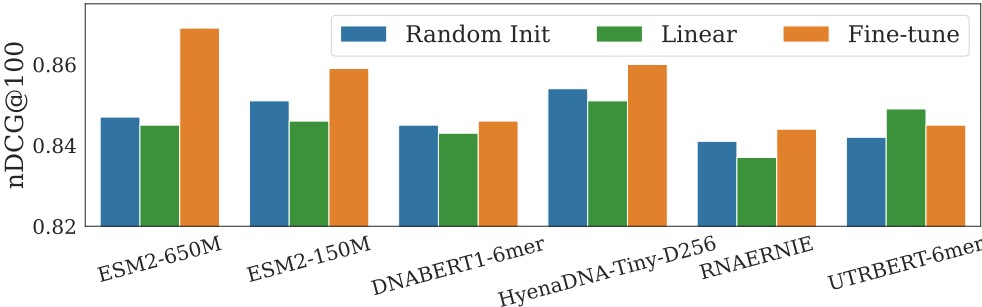

Figure 7: **Biological language models perform poorly on out-of-domain ranking using linear probing and fine-tuning.** We fine-tune the top models in in-domain ranking across each modality, testing a range of learning rates. However, even at the optimal learning rate, the performance remains comparable to that of a randomly initialized ranking head without any training.

from different rounds might fall into distinct domain distributions. Typically, the goal is to predict outcomes of the subsequent round based on results from previous rounds, presenting a classic domain shift problem. Therefore, after exhaustive hyperparameter evolution (as shown in Appendix Table 8), we evaluate the performance of various BLMs on the out-of-domain ranking train-test split based on actual in-vitro evolution rounds, displayed in Figure 7.

We also fine-tune BLMs to ensure that performance limitations are not solely due to the weakness of linear probing. Top models from in-domain ranking across each modality are selected to test at different learning rates. The Table 12 presents the fine-tuning results at the best learning rate. Although fine-tuning improvements a little, most BLMs do not show substantial performance gains. Details of out-of-domain ranking on 80 models across 24 papers are shown in the Appendix Tables 9 to 11. For more analysis such as data precision, please refer to the Appendix A.4.

## 5 CONCLUSION

We present NOVOBENCH-100K, a comprehensive dataset designed to facilitate the in silico evolution of TadA, providing unique insights into base editing. Through extensive benchmarking, we demonstrate that while current biological language models perform well on in-domain ranking, they struggle to generalize effectively on out-of-domain ranking, revealing a significant gap in practical model robustness. These findings highlight the need for developing models capable of handling diverse real-world protein evolution scenarios. Currently, only TadA protein and two rounds of in-vitro evolution are involved in NOVOBENCH-100K. However, we plan to expand the dataset significantly as our wet experiments continue. In a few months, we expect to conduct additional rounds of in vitro evolution and include a broader variety of proteins.

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

# A APPENDIX

## A.1 IN-VITRO EVOLUTION

Phage-Assisted Non-Continuous Evolution (PANCE) represents a sophisticated platform for the directed evolution of biomolecules. This methodology builds upon the principles of Darwinian evolution and leverages the powerful selection capabilities of bacteriophages. The strength of this approach lies in its high-throughput ability to identify the highest-activity protein variants from vast AI-generated starting sequences. In our study, we employed PANCE to evolve TadA, a critical enzyme used in CRISPR base editing, by systematically selecting variants with enhanced activity from vast AI-generated libraries. The convergence of advanced artificial intelligence for library design and PANCE for evolutionary selection represents a frontier in protein engineering, offering a high-throughput and scalable approach to optimize enzymatic functions.

The core of this method is selecting protein variants with improved activity by coupling their function to the replication of bacteriophages. Phages lacking a key gene required for propagation are engineered to rely on the activity of the target protein within host cells to trigger their replication. Through iterative rounds of serial dilution, phages linked to protein variants with higher activity maintain their population, while low-activity counterparts are washed out, allowing for the gradual enrichment of high-performance variants.

We engineered the M13 phage, a filamentous virus that propagates within Escherichia coli (E. coli), to lack the essential gene gIII, which encodes the phage protein pIII, responsible for facilitating the release of new virions from the host. The expression of gIII was made contingent on the activity of TadA within E. coli, such that TadA variants with sufficient activity would trigger gIII expression, enabling phage replication. Each round of PANCE involved the serial dilution of bacterial cultures. Over multiple cycles, variants with superior activity outcompeted their lower-performing counterparts, resulting in a highly refined population of phage-encoded TadA variants. This iterative process ensures that even minimal gains in activity are captured and amplified across generations, gradually evolving TadA to a high-performance state.

A key innovation in our approach is the integration of artificial intelligence to generate a large library of TadA variants before selection. Conventional-directed evolution methods rely heavily on random mutagenesis, which often lacks targeted control and may miss key functional sites, limiting the evolutionary trajectories toward optimal protein variants. In contrast, AI enables the exploration of a much broader sequence space by predicting which mutations are likely to enhance TadA activity based on previously available data and sophisticated machine learning models. This computationally generated library was introduced into the PANCE system, where the selective power of phage was used to identify the highest-performing TadA variants. By coupling AI-driven design with PANCE, we were able to streamline the evolution of TadA toward enhanced activity.

## A.2 SEQ2RANK

Here, we introduce the SEQ2RANK, a novel methodology for processing Next Generation Sequencing (NGS) data by converting sequence-read pairs into a ranking format rather than relying on absolute read counts, which are fraught with risks due to susceptibility to experimental noise and limitations on scalability. We detail the algorithm of SEQ2RANK in Algorithm 1. Our proposed strategy employs an ordering of sequences based on read numbers to represent editing efficiencies, thus pivoting the analytical focus from quantifying individual efficiencies to predicting accurate sequence rankings. This ranking paradigm mitigates challenges inherent in handling vast amounts of NGS data, which include sensitivity issues and the frequent emergence of conflicts arising from biological variability and experimental discrepancies. We utilize a greedy sampling strategy along with a directed acyclic graph (DAG) to maintain a stringent partial order among sequences, thereby ensuring robust data consistency.

## A.3 EVALUATION METHODOLOGY

Normalized Discounted Cumulative Gain (nDCG) is a commonly used metric to evaluate the ranking quality of algorithms, particularly in information retrieval and recommendation systems (Järvelin & Kekäläinen, 2000). It focuses on both the relevance of the ranked items and the position of these

---

**Algorithm 1** Sample Ranking Lists from NGS Data

---

1: **Parameters:** Array of NGS data lists *ngs_lists*, length of ranking list *K*
2: **function** SEQ2RANK(*ngs_lists*, *K*)
3:     Sort *ngs_lists* by the reliability of biological experiments
4:     Initialize directed acyclic graph *G*, sampled ranking lists *results*
5:     **for** *ngs* **in** *ngs_lists* **do**
6:         Build the dictionary of NGS {read: sequences}, *dict_read*
7:         **while** True **do**
8:             $list_K \leftarrow$ GREEDY_SAMPLE(*dict_read*, *G*, *K*, [])
9:             **if** $list_K$ is None **then**
10:                 **Break**
11:             **end if**
12:             Update $list_K$ to *G* and *results*
13:             Remove $list_K$ from *dict_read*
14:         **end while**
15:     **end for**
16:     **return** *results*
17: **end function**
18:
19: **function** GREEDY_SAMPLE(*dict_read*, *G*, *K*, *selected*)
20:     **if** len(*selected*) == *K* **then**
21:         **return** *selected*
22:     **end if**
23:     *seq* ← Find *seq* from *dict_read* which will not introduce cycles in *G*
24:     **if** *seq* is None (cannot find a safe *seq*) **then**
25:         **return** None
26:     **end if**
27:     Append *seq* to *selected*
28:     **return** GREEDY_SAMPLE(*dict_read*, *G*, *K*, *selected*)
29: **end function**

---

items in the ranking list. The relevance score of each item is assigned based on its importance or utility to the user. The gain is discounted logarithmically as the rank increases, meaning that highly relevant items appearing earlier in the ranking list contribute more to the overall score.

The nDCG is normalized by dividing the DCG of the actual ranking by the DCG of the ideal ranking (IDCG), ensuring the score falls within the range of 0 to 1. The DCG (Discounted Cumulative Gain) is calculated as:

$$\text{DCG}_p = \sum_{i=1}^{p} \frac{2^{\text{rel}_i} - 1}{\log_2(i+1)} \tag{1}$$

where $p$ represents the position in the ranking (typically the top $p$ items are evaluated) and $i$ is the rank of the item in the list. The $\text{rel}_i$ is the relevance score of the item at position $i$, which is the reverse ranking in our setting, *i.e.*, the ranking list $1, 2, 3, \ldots$ with a length of $N$ has the $\text{rel}_i$ as $N, N-1, N-2, \ldots$. The $\log_2(i+1)$ is A logarithmic discounting factor that reduces the contribution of lower-ranked items.

The normalized version, nDCG, is calculated as:

$$\text{nDCG}_p = \frac{\text{DCG}_p}{\text{IDCG}_p} \tag{2}$$

where $\text{IDCG}_p$ is the ideal DCG for a perfect ranking.

The nDCG is especially valuable for evaluating ranked retrieval systems because it accounts for the importance of the placement of relevant items within the list. This metric assigns greater weight to items at higher-ranked positions, ensuring that the ranking system's effectiveness is measured more accurately by prioritizing top results, which are typically more relevant to the user. It is particularly suitable for our task of ranking protein activities because we focus more on the top-ranked proteins.

Table 2: **Hyper parameters examinations for in-domain ranking NOVOBENCH-100K @10 task using ESM3 model.** "Random Init" means randomly initializing the model for predictions, essentially guessing the ranking. In the "Base" setting, the learning rate is 0.00001, the batch size is 256, the optimizer is SGD, the hidden size of the linear head module is 256, the training data is shuffled each time, and the loss function used is based on cross-entropy (Cao et al., 2007).

| Hyper Parameter | nDCG↑ | mRR↑ | SP↑ |
|---|---|---|---|
| Random Init | 0.801 | 0.289 | -0.005 |
| Base | 0.856 | 0.333 | 0.202 |
| learning rate = 0.000001 | 0.843 | 0.312 | 0.149 |
| learning rate = 0.0001 | 0.852 | 0.323 | 0.186 |
| learning rate = 0.001 | 0.854 | 0.330 | 0.188 |
| learning rate = 0.01 | 0.811 | 0.292 | 0.029 |
| batch size = 128 | 0.855 | 0.321 | 0.190 |
| batch size = 64 | 0.855 | 0.323 | 0.205 |
| optimizer = Adam | 0.855 | 0.328 | 0.202 |
| optimizer = Adagrad | 0.853 | 0.319 | 0.190 |
| hidden size = 128 | 0.860 | 0.335 | 0.214 |
| hidden size = 512 | 0.858 | 0.326 | 0.207 |
| shuffle = False | 0.857 | 0.329 | 0.205 |
| shuffle = weak shuffle | 0.857 | 0.321 | 0.210 |
| head module = RNN | 0.807 | 0.291 | 0.004 |
| head module = CNN | 0.855 | 0.336 | 0.203 |
| loss func = cosine similarity | 0.812 | 0.285 | 0.019 |
| loss func = kl divergence | 0.798 | 0.292 | -0.017 |

## A.4    IN-DOMAIN PERFORMANCE

For in-domain ranking, we systematically evaluate the hyperparameters in Table 2. We evaluate different learning rates, batch sizes, optimizer types, hidden sizes for the linear head module, whether to keep training data unshuffled, perform a weak shuffle (shuffling only once), shuffle at each training iteration, use different types of head modules, and employ various loss functions.

Begin with the "Base" setting (the learning rate is 0.00001, the batch size is 256, the optimizer is SGD, the hidden size of the linear head module is 256, training data is shuffled each time, and the loss function used is based on cross-entropy (Cao et al., 2007)), we evaluate the impact of various hyperparameter settings on model performance on the NOVOBENCH-100K, including learning rate, batch size, optimizer type, hidden size of the linear head module, data shuffling, head module type, and the loss function.

According to these results, in subsequent benchmarking of various BLMs, we continue with the setting: the hidden size of the linear head module is 128, the optimizer is SGD, and the loss function is cross-entropy based. As for learning rate, considering the varying lengths of embeddings from different BLMs, which result in different sizes for the linear head module, the appropriate learning rate may differ. Therefore, we prepare 3 alternative learning rates for each experiment: 0.001, 0.0001, and 0.00001, selecting the optimal result among the 3 for our analysis. As for batch size, we generally use 64, but for experiments with longer embeddings, we must use smaller batch sizes, such as 16 or 8.

We have also explored the impact of data precision on BLMs in our benchmark by forcibly adjusting the output of the embedding by BLMs to half-precision (*i.e.*, float 16). The resulting nDCG@10, mRR@10, and SP@10 are 0.855, 0.328, and 0.202, respectively. It indicates that using half-precision floating point operations continues to support reasonable performance.

Table 3: **Evaluation on diverse protein BLMs using the linear probing for in-domain ranking.**

| Model | @2 | | | @10 | | | @100 | | |
|---|---|---|---|---|---|---|---|---|---|
| | nDCG↑ | mRR↑ | SP↑ | nDCG↑ | mRR↑ | SP↑ | nDCG↑ | mRR↑ | SP↑ |
| ESM2-8M | 0.828 | 0.767 | 0.069 | 0.836 | 0.321 | 0.136 | 0.874 | 0.053 | 0.176 |
| ESM2-35M | 0.829 | 0.769 | 0.074 | 0.833 | 0.314 | 0.133 | 0.855 | 0.048 | 0.012 |
| ESM2-150M | 0.828 | 0.768 | 0.070 | 0.839 | 0.322 | 0.154 | 0.866 | 0.084 | 0.075 |
| ESM2-650M | 0.830 | 0.770 | 0.080 | 0.840 | 0.324 | 0.162 | 0.875 | 0.075 | 0.138 |
| ESM2-3B | 0.832 | 0.772 | 0.090 | 0.842 | 0.318 | 0.163 | 0.902 | 0.046 | 0.222 |
| ESM2-15B | 0.831 | 0.771 | 0.082 | 0.844 | 0.322 | 0.175 | 0.907 | 0.050 | 0.252 |
| ESM3 | 0.840 | 0.783 | 0.133 | 0.860 | 0.335 | 0.214 | 0.892 | 0.100 | 0.103 |
| SaProt-650M-AF2 | 0.828 | 0.766 | 0.066 | 0.837 | 0.307 | 0.137 | 0.862 | 0.034 | 0.067 |
| SaProt-650M-PDB | 0.831 | 0.771 | 0.083 | 0.839 | 0.322 | 0.144 | 0.864 | 0.042 | 0.085 |
| SaProt-35M-AF2 | 0.832 | 0.773 | 0.091 | 0.835 | 0.311 | 0.134 | 0.877 | 0.069 | 0.144 |
| SaProt-35M-AF2-Seq | 0.830 | 0.769 | 0.077 | 0.837 | 0.323 | 0.143 | 0.870 | 0.058 | 0.066 |
| LucaOne | 0.830 | 0.770 | 0.078 | 0.839 | 0.313 | 0.146 | 0.901 | 0.029 | 0.218 |
| RosettaFold-STATE | 0.823 | 0.760 | 0.040 | 0.817 | 0.299 | 0.058 | 0.851 | 0.078 | 0.010 |
| RosettaFold-MSA | 0.838 | 0.780 | 0.120 | 0.858 | 0.323 | 0.205 | 0.890 | 0.050 | 0.158 |
| ProstT5 | 0.827 | 0.766 | 0.064 | 0.842 | 0.320 | 0.156 | 0.865 | 0.087 | 0.081 |
| ProstT5-fp16 | 0.827 | 0.765 | 0.060 | 0.840 | 0.329 | 0.166 | 0.872 | 0.052 | 0.147 |
| Prot-T5-XL-U50 | 0.832 | 0.773 | 0.090 | 0.835 | 0.320 | 0.144 | 0.880 | 0.053 | 0.160 |
| Prot-T5-XL-Half | 0.834 | 0.775 | 0.102 | 0.835 | 0.309 | 0.143 | 0.868 | 0.048 | 0.098 |
| Chai1 | 0.844 | 0.788 | 0.152 | 0.858 | 0.322 | 0.203 | 0.896 | 0.035 | 0.140 |
| Chai1-ESM | 0.847 | 0.792 | 0.169 | 0.857 | 0.322 | 0.194 | 0.900 | 0.095 | 0.138 |
| Prot-Bert | 0.824 | 0.761 | 0.046 | 0.828 | 0.303 | 0.098 | 0.871 | 0.068 | 0.134 |
| Prot-ss3 | 0.823 | 0.760 | 0.039 | 0.825 | 0.301 | 0.084 | 0.867 | 0.030 | 0.050 |
| Prot-Membrane | 0.830 | 0.770 | 0.079 | 0.829 | 0.316 | 0.119 | 0.862 | 0.062 | 0.028 |
| Prot-Localization | 0.826 | 0.765 | 0.060 | 0.829 | 0.314 | 0.114 | 0.858 | 0.021 | 0.055 |
| Prot-T5-XXL-U50 | 0.832 | 0.773 | 0.090 | 0.842 | 0.320 | 0.177 | 0.882 | 0.045 | 0.154 |
| Prot-Generator | 0.830 | 0.770 | 0.079 | 0.842 | 0.322 | 0.169 | 0.882 | 0.062 | 0.148 |
| Prot-Discriminator | 0.831 | 0.771 | 0.084 | 0.844 | 0.322 | 0.174 | 0.881 | 0.137 | 0.140 |
| Prot-T5-XL-BFD | 0.831 | 0.772 | 0.086 | 0.839 | 0.319 | 0.162 | 0.882 | 0.102 | 0.170 |
| Prot-Bert-BFD | 0.827 | 0.766 | 0.065 | 0.839 | 0.308 | 0.141 | 0.869 | 0.086 | 0.141 |
| Prot-T5-XXL-BFD | 0.831 | 0.771 | 0.085 | 0.844 | 0.325 | 0.172 | 0.886 | 0.038 | 0.185 |
| Prot-Xlnet | 0.830 | 0.769 | 0.077 | 0.833 | 0.314 | 0.141 | 0.880 | 0.060 | 0.110 |
| Prot-Albert | 0.831 | 0.771 | 0.086 | 0.836 | 0.311 | 0.132 | 0.883 | 0.049 | 0.105 |

Then we benchmark 80 models across 24 papers using linear probing, where protein, DNA, and RNA BLMs are shown in Tables 3 to 5 respectively.

When we train the linear head module directly using one-hot vectors of protein sequences, the nDCG@2, nDCG@10, and nDCG@100 results are respectively 0.826, 0.820, and 0.854. Training with one-hot vectors of DNA sequences yields results of 0.819, 0.822, and 0.854; and using one-hot vectors of RNA sequences, the results are 0.819, 0.822, and 0.854. Using embeddings generated by BLMs results in significant improvements. This demonstrates the important value of pre-trained BLMs in downstream applications. We extract the nDCG@100 results from one model in each BLM model family and plot them in the bar chart as shown in the paper main body Figure 5. This visually illustrates that all BLMs outperform results trained solely on sequence one-hot vectors. The performance gap between the 3 modalities of BLMs is not obvious, which means the knowledge of DNA and RNA BLMs is also important in the protein evolution task.

In in-domain ranking experiments, we also observe some certain patterns. First, the scaling law behavior of BLMs is evident. As shown in the paper main body Figure 6, within several classic BLM model families (ESM2, ProtTrans, HyenaDNA, GenSLM, and OmniGenome), as the model parameter scale increases, their performance on our benchmark shows an upward trend, demonstrating a favorable scaling law pattern. Second, we observe that in the 3UTRBERT model family, the performance of the 3-mer model and 6-mer model are better than that of the 4-mer model and 5-mer model. In biological terms, a protein is encoded by exactly three nucleotides, which demonstrates that NOVOBENCH-100K aligns well with the actual biological k-mer patterns.

Table 4: **Evaluation on diverse DNA BLMs using the linear probing for in-domain ranking.**

| Model | @2 | | | @10 | | | @100 | | |
|---|---|---|---|---|---|---|---|---|---|
| | nDCG↑ | mRR↑ | SP↑ | nDCG↑ | mRR↑ | SP↑ | nDCG↑ | mRR↑ | SP↑ |
| EVO-8k | 0.829 | 0.769 | 0.075 | 0.851 | 0.308 | 0.183 | 0.891 | 0.045 | 0.153 |
| EVO-131k | 0.830 | 0.770 | 0.080 | 0.850 | 0.317 | 0.190 | 0.895 | 0.007 | 0.153 |
| LucaOne | 0.835 | 0.776 | 0.106 | 0.843 | 0.303 | 0.165 | 0.888 | 0.037 | 0.165 |
| Chai1 | 0.848 | 0.794 | 0.175 | 0.868 | 0.322 | 0.224 | 0.901 | 0.086 | 0.222 |
| NT-2-50M | 0.833 | 0.774 | 0.097 | 0.845 | 0.305 | 0.179 | 0.879 | 0.057 | 0.137 |
| NT-2-100M | 0.836 | 0.777 | 0.109 | 0.843 | 0.306 | 0.171 | 0.872 | 0.068 | 0.118 |
| NT-2-250M | 0.830 | 0.770 | 0.078 | 0.845 | 0.312 | 0.171 | 0.866 | 0.081 | 0.088 |
| NT-2-500M | 0.834 | 0.776 | 0.102 | 0.849 | 0.313 | 0.202 | 0.880 | 0.027 | 0.176 |
| NT-500M-human-ref | 0.836 | 0.777 | 0.109 | 0.840 | 0.292 | 0.154 | 0.892 | 0.098 | 0.195 |
| NT-500M-1000G | 0.833 | 0.774 | 0.097 | 0.847 | 0.314 | 0.191 | 0.864 | 0.034 | 0.107 |
| NT-2B5-1000G | 0.836 | 0.777 | 0.109 | 0.845 | 0.307 | 0.180 | 0.884 | 0.030 | 0.182 |
| NT-2B5-multi-species | 0.828 | 0.767 | 0.069 | 0.838 | 0.291 | 0.145 | 0.872 | 0.035 | 0.110 |
| AgroNT | 0.831 | 0.772 | 0.086 | 0.839 | 0.304 | 0.156 | 0.868 | 0.096 | 0.123 |
| GenSLMs 2.5B | 0.836 | 0.777 | 0.109 | 0.857 | 0.327 | 0.204 | 0.904 | 0.043 | 0.233 |
| GenSLMs 250M | 0.836 | 0.777 | 0.109 | 0.856 | 0.326 | 0.204 | 0.907 | 0.040 | 0.249 |
| GenSLMs 25M | 0.831 | 0.771 | 0.084 | 0.837 | 0.322 | 0.160 | 0.892 | 0.072 | 0.178 |
| DNABERT-2-117M | 0.816 | 0.750 | 0.001 | 0.814 | 0.295 | 0.038 | 0.861 | 0.026 | 0.018 |
| DNABERT-S | 0.817 | 0.752 | 0.007 | 0.812 | 0.301 | 0.028 | 0.853 | 0.040 | 0.017 |
| DNABERT-1-3mer | 0.830 | 0.770 | 0.081 | 0.841 | 0.303 | 0.163 | 0.879 | 0.144 | 0.085 |
| DNABERT-1-4mer | 0.830 | 0.770 | 0.080 | 0.836 | 0.299 | 0.138 | 0.872 | 0.043 | 0.089 |
| DNABERT-1-5mer | 0.837 | 0.779 | 0.114 | 0.849 | 0.313 | 0.179 | 0.874 | 0.043 | 0.142 |
| DNABERT-1-6mer | 0.835 | 0.776 | 0.105 | 0.845 | 0.299 | 0.163 | 0.893 | 0.075 | 0.236 |
| HyenaDNA-T | 0.832 | 0.773 | 0.092 | 0.844 | 0.314 | 0.178 | 0.864 | 0.032 | 0.037 |
| HyenaDNA-T-d256 | 0.835 | 0.776 | 0.104 | 0.848 | 0.325 | 0.195 | 0.886 | 0.043 | 0.200 |
| HyenaDNA-T-d128 | 0.830 | 0.770 | 0.079 | 0.843 | 0.313 | 0.166 | 0.864 | 0.025 | 0.112 |
| HyenaDNA-S | 0.830 | 0.770 | 0.081 | 0.842 | 0.306 | 0.177 | 0.870 | 0.069 | 0.098 |
| HyenaDNA-M-160k | 0.831 | 0.771 | 0.083 | 0.848 | 0.314 | 0.186 | 0.884 | 0.037 | 0.115 |
| HyenaDNA-M-450k | 0.832 | 0.772 | 0.089 | 0.845 | 0.309 | 0.172 | 0.873 | 0.064 | 0.113 |
| HyenaDNA-L | 0.831 | 0.771 | 0.085 | 0.848 | 0.314 | 0.178 | 0.883 | 0.029 | 0.132 |

Table 5: **Evaluation on diverse RNA BLMs using the linear probing for in-domain ranking.**

| Model | @2 | | | @10 | | | @100 | | |
|---|---|---|---|---|---|---|---|---|---|
| | nDCG↑ | mRR↑ | SP↑ | nDCG↑ | mRR↑ | SP↑ | nDCG↑ | mRR↑ | SP↑ |
| mRNA-FM | 0.830 | 0.770 | 0.079 | 0.846 | 0.315 | 0.187 | 0.880 | 0.026 | 0.117 |
| RNA-FM | 0.831 | 0.771 | 0.084 | 0.838 | 0.293 | 0.146 | 0.879 | 0.026 | 0.161 |
| RNA-MSM | 0.832 | 0.773 | 0.090 | 0.850 | 0.317 | 0.197 | 0.902 | 0.045 | 0.253 |
| RNA-Ernie | 0.837 | 0.780 | 0.119 | 0.867 | 0.326 | 0.230 | 0.906 | 0.056 | 0.237 |
| RiNaLMo | 0.843 | 0.787 | 0.148 | 0.870 | 0.326 | 0.235 | 0.904 | 0.049 | 0.198 |
| ERNIERNA | 0.836 | 0.778 | 0.113 | 0.860 | 0.327 | 0.230 | 0.909 | 0.041 | 0.213 |
| ERNIERNA.ss | 0.837 | 0.779 | 0.115 | 0.860 | 0.323 | 0.226 | 0.906 | 0.031 | 0.234 |
| Chai1 | 0.845 | 0.790 | 0.161 | 0.867 | 0.316 | 0.225 | 0.897 | 0.124 | 0.217 |
| OmniGenome-418M | 0.838 | 0.781 | 0.122 | 0.868 | 0.320 | 0.239 | 0.910 | 0.059 | 0.207 |
| OmniGenome-186M | 0.839 | 0.782 | 0.127 | 0.861 | 0.313 | 0.210 | 0.896 | 0.061 | 0.213 |
| OmniGenome-52M | 0.831 | 0.771 | 0.083 | 0.846 | 0.327 | 0.184 | 0.886 | 0.041 | 0.207 |
| 3UTRBERT-6mer | 0.840 | 0.784 | 0.135 | 0.870 | 0.324 | 0.244 | 0.908 | 0.044 | 0.256 |
| 3UTRBERT-5mer | 0.834 | 0.775 | 0.102 | 0.861 | 0.323 | 0.231 | 0.906 | 0.046 | 0.232 |
| 3UTRBERT-4mer | 0.840 | 0.784 | 0.134 | 0.869 | 0.326 | 0.243 | 0.906 | 0.047 | 0.234 |
| 3UTRBERT-3mer | 0.841 | 0.785 | 0.138 | 0.870 | 0.323 | 0.246 | 0.906 | 0.041 | 0.226 |
| SpliceBERT | 0.829 | 0.768 | 0.072 | 0.836 | 0.307 | 0.140 | 0.872 | 0.058 | 0.114 |
| SpliceBERT-H.510nt | 0.833 | 0.774 | 0.095 | 0.844 | 0.308 | 0.167 | 0.890 | 0.051 | 0.203 |
| SpliceBERT.510nt | 0.830 | 0.770 | 0.080 | 0.838 | 0.310 | 0.152 | 0.874 | 0.036 | 0.121 |
| CaLM | 0.834 | 0.775 | 0.099 | 0.847 | 0.309 | 0.178 | 0.882 | 0.062 | 0.146 |

Table 6: **BLMs perform well on in-domain ranking using linear probing and fine-tuning.** The table shows the result of @100 track with a batch size of $1 \times 100$ sequences. For most selected models except 3UTRBERT-6mer, fine-tuning provides better results than linear probing.

| Modality | Model | Random Initialization | | Linear Probing | | Fine-tuning | |
|---|---|---|---|---|---|---|---|
| | | nDCG↑ | SP↑ | nDCG↑ | SP↑ | nDCG↑ | SP↑ |
| Protein | ESM2-650M | 0.844 | -0.050 | 0.875 | 0.138 | 0.902 | 0.208 |
| | ESM2-150M | 0.856 | 0.050 | 0.866 | 0.075 | 0.898 | 0.187 |
| DNA | DNABERT-1-6mer | 0.856 | 0.017 | 0.893 | 0.236 | 0.908 | 0.226 |
| | HyenaDNA-T-d256 | 0.865 | 0.053 | 0.886 | 0.200 | 0.908 | 0.205 |
| RNA | RNA-Ernie | 0.855 | 0.003 | 0.906 | 0.237 | 0.907 | 0.239 |
| | 3UTRBERT-6mer | 0.836 | -0.015 | 0.908 | 0.256 | 0.902 | 0.210 |

Also, we report fine-tuning performance on in-domain ranking, shown in Table 6. Firstly, linear probing and fine-tuning effectively surpass the random init in nDCG@100 and SP. Secondly, fine-tuning provides better results than linear probing for most selected models except 3UTRBERT-6mer. Thirdly, SP and n@DCG can provide different tendencies, demonstrating the different concentrations for distinct metrics, shown in Section 4.1.2. For example, the nDCG of linear probing in 3UTRBERT-6mer is higher than fine-tuning, while the SP is the opposite. The linear probing performs better at the top sequences, while the fine-tuning shows better rankings on 100 sequences.

## A.5 OUT-OF-DOMAIN PERFORMANCE

The train-test splitting of out-of-domain ranking is shown in Table 7. We emphasize that although the out-of-domain ranking is a challenging task, it aligns with the logic of the actual process of protein evolution.

Table 7: **The out-of-domain ranking is highly challenging as it is based on actual in-vitro evolution rounds.** We provide three tracks, @2, @10, and @100, where the lengths of ranking lists are 2, 10, and 100 respectively.

| Track | #List | | #DNA | | #Protein | |
|---|---|---|---|---|---|---|
| | Train | Test | Train | Test | Train | Test |
| @100 | 7 | 99 | 682 | 9822 | 661 | 9159 |
| @10 | 1155 | 4563 | 8745 | 41264 | 5398 | 24906 |
| @2 | 27754 | 44322 | 38114 | 63445 | 16461 | 28800 |

For out-of-domain ranking, we conduct a comprehensive evaluation of various hyperparameters, as detailed in Table 8. We assess the impact of different learning rates, batch sizes, and optimizer types to understand their influence on model performance. Additionally, we explore various configurations for the linear head module, including different hidden sizes, to determine how these settings affect the results.

Another important aspect of our evaluation is the data shuffling strategy. We experiment with three approaches: leaving the training data unshuffled, performing a weak shuffle by shuffling only once before training, and shuffling the data at each training iteration. The goal is to identify whether data presented during training influences model generalization and performance.

Furthermore, we test different head module types to find the most effective architecture for the task. Alongside this, we explore a variety of loss functions to optimize the model for ranking. Despite the thorough tuning of these hyperparameters and the exploration of different settings, all configurations result in disappointing evaluation outcomes.

Table 8: **Hyper parameters examinations for out-of-domain ranking NOVOBENCH-100K @10 task using ESM3 model.** "Random Init" means randomly initializing the model for predictions, essentially guessing the ranking. In the "Base" setting, the learning rate is 0.00001, the batch size is 256, the optimizer is SGD, the hidden size of the linear head module is 256, the training data is shuffled each time, and the loss function used is based on cross-entropy (Cao et al., 2007). No matter how we tune the hyperparameters, the linear probe performs poorly.

| Hyper Parameter | nDCG↑ | mRR↑ | SP↑ |
|---|---|---|---|
| Random Init | 0.803 | 0.301 | -0.007 |
| Base | 0.806 | 0.294 | 0.014 |
| learning rate = 0.000001 | 0.801 | 0.297 | -0.010 |
| learning rate = 0.0001 | 0.807 | 0.290 | 0.011 |
| learning rate = 0.001 | 0.813 | 0.301 | 0.033 |
| learning rate = 0.01 | 0.803 | 0.293 | -0.012 |
| batch size = 128 | 0.811 | 0.301 | 0.030 |
| batch size = 64 | 0.810 | 0.298 | 0.030 |
| optimizer = Adam | 0.810 | 0.301 | 0.024 |
| optimizer = Adagrad | 0.812 | 0.293 | 0.032 |
| hidden size = 128 | 0.809 | 0.289 | 0.018 |
| hidden size = 512 | 0.810 | 0.300 | 0.027 |
| shuffle = False | 0.806 | 0.290 | 0.007 |
| shuffle = weak shuffle | 0.808 | 0.301 | 0.022 |
| head module = RNN | 0.809 | 0.309 | 0.028 |
| head module = CNN | 0.805 | 0.294 | 0.008 |
| loss func = cosine similarity | 0.796 | 0.288 | -0.041 |
| loss func = kl divergence | 0.796 | 0.292 | -0.030 |

These results suggest that the challenges of out-of-domain ranking are not easily addressed through standard hyperparameter adjustments. It indicates a need for more advanced strategies beyond conventional tuning to improve the model's performance in out-of-domain ranking.

We proceed to benchmark all 80 biological language models (BLMs) reported in 24 papers using a linear probing approach. The results are presented separately for protein, DNA, and RNA BLMs in Table 9, Table 10, and Table 11, respectively. Across all modalities, most BLMs perform poorly on out-of-domain ranking, with results barely surpassing those of random guess ranking.

This poor performance stands in stark contrast to the outcomes observed in in-domain ranking, where nearly all BLMs achieve results consistent with expectations, as shown in Table 3, Table 4, and Table 5. These results confirm that the embeddings generated by BLMs are meaningful and effective in in-domain tasks, demonstrating no apparent issues related to the curse of dimensionality or loss of information during the embedding process.

The disparity between in-domain ranking and out-of-domain ranking performance suggests that the challenges faced by BLMs in out-of-domain ranking are not due to the embeddings themselves but are likely attributed to the difficulty of generalizing to out-of-domain data. While the embeddings remain useful within the context of in-domain ranking tasks, their transferability and robustness across varying experimental conditions in out-of-domain ranking are limited. This emphasizes the need for more advanced strategies to enhance the generalization ability of BLMs when faced with out-of-domain ranking tasks.

This underscores the significance of developing more robust approaches to improve the generalizability of BLMs. Given that the embeddings are effective for in-domain tasks but struggle with out-of-domain scenarios, it becomes crucial to explore new strategies beyond traditional training and fine-tuning. Potential directions include leveraging multimodal data, incorporating external biological

Table 9: **Protein BLMs fail to solve the out-of-domain ranking using linear probing.**

| Model | @2 | | | @10 | | | @100 | | |
|---|---|---|---|---|---|---|---|---|---|
| | nDCG↑ | mRR↑ | SP↑ | nDCG↑ | mRR↑ | SP↑ | nDCG↑ | mRR↑ | SP↑ |
| ESM2-8M | 0.811 | 0.744 | -0.023 | 0.815 | 0.310 | 0.061 | 0.856 | 0.053 | 0.014 |
| ESM2-35M | 0.810 | 0.743 | -0.028 | 0.803 | 0.298 | -0.005 | 0.858 | 0.055 | 0.060 |
| ESM2-150M | 0.811 | 0.744 | -0.024 | 0.808 | 0.293 | 0.023 | 0.856 | 0.057 | 0.028 |
| ESM2-650M | 0.814 | 0.747 | -0.010 | 0.802 | 0.293 | -0.004 | 0.845 | 0.049 | 0.016 |
| ESM2-3B | 0.815 | 0.750 | -0.001 | 0.808 | 0.308 | 0.027 | 0.838 | 0.037 | -0.018 |
| ESM2-15B | 0.815 | 0.749 | -0.002 | 0.801 | 0.300 | 0.000 | 0.834 | 0.064 | -0.071 |
| ESM3 | 0.819 | 0.755 | 0.018 | 0.802 | 0.298 | -0.004 | 0.864 | 0.054 | 0.069 |
| SaProt-650M-AF2 | 0.813 | 0.747 | -0.011 | 0.797 | 0.284 | -0.036 | 0.853 | 0.055 | 0.060 |
| SaProt-650M-PDB | 0.817 | 0.752 | 0.006 | 0.802 | 0.294 | -0.009 | 0.836 | 0.063 | -0.046 |
| SaProt-35M-AF2 | 0.817 | 0.751 | 0.006 | 0.812 | 0.306 | 0.046 | 0.845 | 0.057 | -0.029 |
| SaProt-35M-AF2-Seq | 0.821 | 0.757 | 0.029 | 0.802 | 0.297 | -0.016 | 0.854 | 0.079 | 0.025 |
| LucaOne | 0.823 | 0.761 | 0.044 | 0.798 | 0.291 | -0.021 | 0.846 | 0.069 | 0.007 |
| RosettaFold-STATE | 0.812 | 0.746 | -0.017 | 0.801 | 0.282 | -0.018 | 0.837 | 0.044 | -0.055 |
| RosettaFold-MSA | 0.811 | 0.745 | -0.022 | 0.800 | 0.285 | -0.022 | 0.844 | 0.077 | 0.001 |
| ProstT5 | 0.817 | 0.752 | 0.006 | 0.804 | 0.289 | -0.004 | 0.843 | 0.041 | -0.038 |
| ProstT5-fp16 | 0.816 | 0.750 | 0.002 | 0.801 | 0.300 | -0.006 | 0.852 | 0.043 | 0.012 |
| Prot-T5-XL-U50 | 0.815 | 0.749 | -0.003 | 0.807 | 0.307 | 0.019 | 0.852 | 0.053 | 0.040 |
| Prot-T5-XL-Half | 0.810 | 0.742 | -0.031 | 0.803 | 0.287 | -0.010 | 0.855 | 0.038 | 0.008 |
| Chai1 | 0.814 | 0.748 | -0.008 | 0.802 | 0.290 | -0.008 | 0.857 | 0.055 | 0.062 |
| Chai1-ESM | 0.808 | 0.740 | -0.042 | 0.803 | 0.296 | -0.008 | 0.842 | 0.033 | -0.050 |
| Prot-Bert | 0.819 | 0.755 | 0.021 | 0.817 | 0.304 | 0.059 | 0.843 | 0.047 | -0.024 |
| Prot-ss3 | 0.813 | 0.747 | -0.012 | 0.804 | 0.290 | 0.000 | 0.840 | 0.047 | -0.035 |
| Prot-Membrane | 0.822 | 0.758 | 0.033 | 0.805 | 0.298 | -0.001 | 0.853 | 0.067 | 0.035 |
| Prot-Localization | 0.807 | 0.738 | -0.048 | 0.802 | 0.296 | -0.005 | 0.841 | 0.050 | -0.033 |
| Prot-T5-XXL-U50 | 0.816 | 0.751 | 0.003 | 0.799 | 0.305 | -0.020 | 0.857 | 0.062 | 0.039 |
| Prot-Generator | 0.818 | 0.754 | 0.015 | 0.804 | 0.301 | 0.003 | 0.849 | 0.073 | 0.013 |
| Prot-Discriminator | 0.816 | 0.750 | 0.000 | 0.801 | 0.290 | -0.090 | 0.851 | 0.051 | -0.005 |
| Prot-T5-XL-BFD | 0.815 | 0.750 | 0.000 | 0.799 | 0.297 | -0.016 | 0.844 | 0.069 | -0.012 |
| Prot-Bert-BFD | 0.814 | 0.748 | -0.010 | 0.803 | 0.291 | -0.001 | 0.837 | 0.039 | -0.044 |
| Prot-T5-XXL-BFD | 0.813 | 0.746 | -0.015 | 0.808 | 0.299 | 0.024 | 0.841 | 0.049 | -0.025 |
| Prot-Xlnet | 0.813 | 0.747 | -0.012 | 0.803 | 0.292 | 0.004 | 0.839 | 0.054 | 0.003 |
| Prot-Albert | 0.812 | 0.745 | -0.020 | 0.796 | 0.288 | -0.037 | 0.838 | 0.044 | -0.036 |

knowledge to better inform model predictions, or applying advanced domain adaptation techniques specifically tailored to biological contexts.

Moreover, improving model architecture and pre-training processes may also play a role in enhancing the ability of BLMs to generalize. For example, introducing mechanisms to better capture contextual dependencies across sequences or developing models that can dynamically adapt to new types of biological data could mitigate the current performance gaps. These improvements could ultimately address the limitations seen in out-of-domain ranking and facilitate more accurate predictions in real-world applications where models frequently encounter data that differs from the training distribution.

In summary, while current BLMs perform well on in-domain ranking tasks, their poor performance on out-of-domain tasks points to a pressing need for methodological advances that enable consistent generalization across varying biological experiments. This remains a critical step toward fully realizing the potential of BLMs in supporting accurate and practical biological research.

Table 10: **DNA BLMs fail to solve the out-of-domain ranking using linear probing.**

| Model | @2 | | | @10 | | | @100 | | |
|---|---|---|---|---|---|---|---|---|---|
| | nDCG↑ | mRR↑ | SP↑ | nDCG↑ | mRR↑ | SP↑ | nDCG↑ | mRR↑ | SP↑ |
| EVO-8k | 0.809 | 0.741 | -0.036 | 0.799 | 0.286 | -0.022 | 0.831 | 0.043 | -0.079 |
| EVO-131k | 0.809 | 0.741 | -0.037 | 0.802 | 0.293 | -0.016 | 0.833 | 0.054 | -0.080 |
| LucaOne | 0.816 | 0.750 | 0.001 | 0.808 | 0.289 | 0.006 | 0.839 | 0.055 | 0.013 |
| Chai1 | 0.820 | 0.756 | 0.025 | 0.802 | 0.292 | -0.015 | 0.851 | 0.063 | 0.009 |
| NT-2-50M | 0.812 | 0.746 | -0.017 | 0.802 | 0.291 | -0.019 | 0.837 | 0.035 | -0.025 |
| NT-2-100M | 0.818 | 0.753 | 0.011 | 0.800 | 0.290 | -0.024 | 0.857 | 0.070 | 0.052 |
| NT-2-250M | 0.818 | 0.753 | 0.013 | 0.805 | 0.288 | 0.004 | 0.849 | 0.037 | 0.007 |
| NT-2-500M | 0.816 | 0.751 | 0.005 | 0.804 | 0.289 | -0.013 | 0.843 | 0.047 | -0.044 |
| NT-500M-human | 0.812 | 0.745 | -0.021 | 0.806 | 0.291 | 0.005 | 0.829 | 0.051 | -0.108 |
| NT-500M-1000G | 0.816 | 0.751 | 0.004 | 0.804 | 0.295 | 0.001 | 0.841 | 0.059 | -0.025 |
| NT-2B5-1000G | 0.815 | 0.749 | -0.003 | 0.805 | 0.301 | 0.005 | 0.856 | 0.034 | 0.046 |
| NT-2B5 | 0.820 | 0.757 | 0.027 | 0.803 | 0.297 | -0.008 | 0.840 | 0.026 | -0.033 |
| AgroNT | 0.815 | 0.749 | -0.003 | 0.815 | 0.298 | 0.038 | 0.830 | 0.056 | -0.083 |
| GenSLMs-2.5B | 0.810 | 0.743 | -0.029 | 0.810 | 0.300 | 0.027 | 0.857 | 0.043 | 0.066 |
| GenSLMs-250M | 0.812 | 0.746 | -0.018 | 0.799 | 0.289 | -0.028 | 0.853 | 0.049 | 0.020 |
| GenSLMs-25M | 0.819 | 0.755 | 0.020 | 0.807 | 0.298 | 0.007 | 0.841 | 0.050 | 0.002 |
| DNABERT-2 | 0.813 | 0.747 | -0.015 | 0.802 | 0.285 | -0.024 | 0.863 | 0.062 | 0.072 |
| DNABERT-S | 0.812 | 0.745 | -0.019 | 0.801 | 0.288 | -0.026 | 0.851 | 0.043 | 0.036 |
| DNABERT1-3mer | 0.818 | 0.753 | 0.014 | 0.801 | 0.289 | -0.023 | 0.851 | 0.041 | 0.039 |
| DNABERT1-4mer | 0.815 | 0.749 | -0.002 | 0.804 | 0.288 | -0.007 | 0.840 | 0.043 | -0.026 |
| DNABERT1-5mer | 0.818 | 0.753 | 0.011 | 0.806 | 0.297 | 0.007 | 0.850 | 0.062 | -0.001 |
| DNABERT1-6mer | 0.811 | 0.744 | -0.025 | 0.809 | 0.296 | 0.019 | 0.843 | 0.060 | -0.049 |
| HyenaDNA-T | 0.816 | 0.751 | 0.004 | 0.808 | 0.294 | 0.006 | 0.828 | 0.046 | -0.124 |
| HyenaDNA-T-d128 | 0.817 | 0.753 | 0.011 | 0.800 | 0.286 | -0.044 | 0.845 | 0.039 | -0.024 |
| HyenaDNA-T-d256 | 0.816 | 0.750 | 0.002 | 0.803 | 0.286 | -0.007 | 0.851 | 0.038 | 0.029 |
| HyenaDNA-S | 0.817 | 0.752 | 0.006 | 0.816 | 0.292 | 0.047 | 0.857 | 0.076 | 0.027 |
| HyenaDNA-M-160k | 0.817 | 0.752 | 0.010 | 0.800 | 0.282 | -0.023 | 0.850 | 0.042 | -0.003 |
| HyenaDNA-M-450k | 0.819 | 0.755 | 0.021 | 0.803 | 0.284 | -0.027 | 0.844 | 0.060 | -0.052 |
| HyenaDNA-L | 0.814 | 0.748 | -0.008 | 0.804 | 0.288 | -0.019 | 0.861 | 0.045 | 0.061 |

Table 11: **RNA BLMs fail to solve the out-of-domain ranking using linear probing.**

| Model | @2 | | | @10 | | | @100 | | |
|---|---|---|---|---|---|---|---|---|---|
| | nDCG↑ | mRR↑ | SP↑ | nDCG↑ | mRR↑ | SP↑ | nDCG↑ | mRR↑ | SP↑ |
| mRNA-FM | 0.814 | 0.748 | -0.006 | 0.809 | 0.291 | 0.015 | 0.847 | 0.045 | 0.003 |
| RNA-FM | 0.813 | 0.747 | -0.013 | 0.814 | 0.303 | 0.045 | 0.852 | 0.037 | 0.020 |
| RNA-MSM | 0.821 | 0.757 | 0.029 | 0.811 | 0.299 | 0.021 | 0.844 | 0.060 | 0.007 |
| RNA-Ernie | 0.815 | 0.750 | -0.001 | 0.803 | 0.298 | -0.007 | 0.837 | 0.056 | -0.039 |
| RiNALMo | 0.817 | 0.751 | 0.006 | 0.807 | 0.291 | 0.017 | 0.832 | 0.046 | -0.037 |
| ERNIE-RNA | 0.816 | 0.750 | 0.001 | 0.803 | 0.293 | -0.010 | 0.853 | 0.065 | 0.050 |
| ERNIE-RNA.ss | 0.817 | 0.751 | 0.006 | 0.807 | 0.296 | 0.007 | 0.864 | 0.076 | 0.070 |
| Chai1 | 0.814 | 0.748 | -0.006 | 0.809 | 0.293 | 0.018 | 0.853 | 0.075 | 0.044 |
| OmniGenome-418M | 0.817 | 0.752 | 0.009 | 0.799 | 0.287 | -0.037 | 0.840 | 0.042 | -0.017 |
| OmniGenome-186M | 0.819 | 0.755 | 0.019 | 0.810 | 0.297 | 0.017 | 0.842 | 0.088 | -0.036 |
| OmniGenome-52M | 0.814 | 0.749 | -0.006 | 0.812 | 0.296 | 0.036 | 0.835 | 0.076 | -0.040 |
| 3UTRBERT-6mer | 0.812 | 0.745 | -0.019 | 0.811 | 0.293 | 0.029 | 0.849 | 0.074 | 0.025 |
| 3UTRBERT-5mer | 0.819 | 0.755 | 0.020 | 0.811 | 0.293 | 0.026 | 0.842 | 0.044 | 0.028 |
| 3UTRBERT-4mer | 0.820 | 0.756 | 0.024 | 0.800 | 0.285 | -0.029 | 0.850 | 0.055 | 0.017 |
| 3UTRBERT-3mer | 0.815 | 0.750 | 0.000 | 0.809 | 0.299 | 0.299 | 0.842 | 0.045 | -0.056 |
| SpliceBERT | 0.814 | 0.748 | -0.008 | 0.802 | 0.298 | -0.011 | 0.856 | 0.049 | 0.035 |
| SpliceBERT-H.510nt | 0.814 | 0.748 | -0.008 | 0.805 | 0.297 | 0.004 | 0.839 | 0.045 | -0.077 |
| SpliceBERT.510nt | 0.817 | 0.752 | 0.007 | 0.801 | 0.294 | -0.015 | 0.847 | 0.065 | -0.005 |
| CaLM | 0.817 | 0.752 | 0.009 | 0.800 | 0.285 | -0.031 | 0.840 | 0.080 | -0.056 |

Table 12: **BLMs struggle to solve out-of-domain ranking using linear probing and fine-tuning.** The table shows the result of @100 track with a batch size of $4 \times 100$ sequences. Although fine-tuning can help a little, most BLMs cannot solve the out-of-domain ranking well with a similar performance of random initialization.

| Modality | Model | Random Initialization | | Linear Probing | | Fine-tuning | |
|---|---|---|---|---|---|---|---|
| | | nDCG↑ | SP↑ | nDCG↑ | SP↑ | nDCG↑ | SP↑ |
| Protein | ESM2-650M | 0.847 | 0.017 | 0.845 | 0.016 | 0.869 | 0.114 |
| | ESM2-150M | 0.851 | 0.010 | 0.846 | 0.007 | 0.859 | 0.051 |
| DNA | DNABERT-1-6mer | 0.845 | -0.057 | 0.843 | -0.049 | 0.846 | 0.009 |
| | HyenaDNA-T-d256 | 0.854 | 0.018 | 0.851 | 0.029 | 0.860 | 0.068 |
| RNA | RNAErnie | 0.841 | 0.007 | 0.837 | -0.039 | 0.844 | -0.003 |
| | 3UTRBERT-6mer | 0.842 | -0.042 | 0.849 | 0.025 | 0.845 | 0.007 |

We conduct fine-tuning experiments in Table 12, training the BLM backbones and their ranking heads to ensure that performance limitations are not solely due to linear probing. We fine-tune the top models from in-domain ranking across each modality, testing different learning rates. Table 12 presents the fine-tuning results at the best learning rate. Although fine-tuning provides improvements over the random init, most BLMs do not show substantial performance gains. This indicates that when BLMs face out-of-domain ranking tasks in our benchmark, i.e., predicting the outcomes of the next round of protein evolution based on results from the current round, they are almost incapable. This reflects the considerable challenge posed by our benchmark in out-of-domain ranking tasks with existing BLMs. Such challenges align with the logic of actual biological experiments and represent real difficulties that need resolution in practical applications.

We have also studied the performance by data precision in Table 13, which is one of the common tasks for studying language model performance. In the task of out-of-domain ranking, even using half the accuracy does not cause the model to crash on our benchmark.

Table 13: **Evaluation on diverse data precisions using linear probing (out-of-domain ranking task).** The top, middle, and bottom blocks represent the protein, DNA, and RNA modalities respectively. "P" represents the precision.

| Model | P | @2 | | | @10 | | | @100 | | |
|---|---|---|---|---|---|---|---|---|---|---|
| | | nDCG↑ | mRR↑ | SP↑ | nDCG↑ | mRR↑ | SP↑ | nDCG↑ | mRR↑ | SP↑ |
| ESM2-650M | F32 | 0.814 | 0.747 | -0.010 | 0.802 | 0.293 | -0.004 | 0.845 | 0.049 | 0.016 |
| | F16 | 0.817 | 0.752 | 0.011 | 0.803 | 0.291 | 0.002 | 0.841 | 0.045 | -0.010 |
| ESM3 | F32 | 0.819 | 0.755 | 0.018 | 0.802 | 0.298 | -0.004 | 0.864 | 0.054 | 0.069 |
| | F16 | 0.815 | 0.749 | -0.001 | 0.804 | 0.295 | -0.001 | 0.849 | 0.070 | -0.013 |
| LucaOne | F32 | 0.823 | 0.761 | 0.044 | 0.798 | 0.291 | -0.021 | 0.846 | 0.069 | 0.007 |
| | F16 | 0.816 | 0.750 | 0.005 | 0.812 | 0.294 | 0.042 | 0.836 | 0.050 | -0.008 |
| HyenaDNA-L | F32 | 0.814 | 0.748 | -0.008 | 0.804 | 0.288 | -0.019 | 0.861 | 0.045 | 0.061 |
| | F16 | 0.814 | 0.747 | -0.008 | 0.794 | 0.289 | -0.055 | 0.857 | 0.048 | 0.065 |
| EVO-131k | F32 | 0.809 | 0.741 | -0.037 | 0.802 | 0.293 | -0.016 | 0.833 | 0.054 | -0.080 |
| | F16 | 0.813 | 0.744 | -0.012 | 0.808 | 0.297 | 0.026 | 0.846 | 0.048 | 0.013 |
| LucaOne† | F32 | 0.816 | 0.750 | 0.001 | 0.808 | 0.289 | 0.006 | 0.839 | 0.055 | 0.013 |
| | F16 | 0.816 | 0.747 | 0.003 | 0.810 | 0.300 | 0.025 | 0.839 | 0.058 | -0.006 |
| SpliceBERT | F32 | 0.814 | 0.748 | -0.008 | 0.802 | 0.298 | -0.011 | 0.856 | 0.049 | 0.035 |
| | F16 | 0.818 | 0.752 | 0.013 | 0.812 | 0.289 | 0.024 | 0.845 | 0.048 | -0.039 |
| 3UTRBERT-6mer | F32 | 0.812 | 0.745 | -0.019 | 0.811 | 0.293 | 0.029 | 0.849 | 0.074 | 0.025 |
| | F16 | 0.820 | 0.755 | 0.022 | 0.803 | 0.280 | -0.013 | 0.863 | 0.040 | 0.032 |
| OmniGenome-52M | F32 | 0.814 | 0.749 | -0.006 | 0.812 | 0.296 | 0.036 | 0.835 | 0.076 | -0.040 |
| | F16 | 0.814 | 0.748 | -0.007 | 0.805 | 0.298 | 0.007 | 0.855 | 0.060 | 0.059 |

