# OpenReview forum: "NovoBench-100K: A large-scale protein dataset for in silico evolution of de novo TadA"
_ICLR.cc/2025/Conference — Submitted to ICLR 2025_

### Official Review · Reviewer_ejN3 · 2024-10-31

**Soundness:** 3
**Presentation:** 2
**Contribution:** 2
**Rating:** 5
**Confidence:** 4

**Summary:**

While numerous datasets exist to evaluate model effectiveness in predicting protein structures, this paper highlights the need for datasets focused on assessing model performance in predicting proteins with specific functional capabilities. For example, can models predict functional variants of TadA, where certain high-efficiency variants demonstrate superior gene-editing capabilities compared to control counterparts. To address this, the authors introduce the dataset NovoBench 100K, which includes ~100 unique DNA variants, each experimentally derived by the authors in a wet lab and associated with a value indicating editing efficiency.

This dataset is divided to provide both in-domain and out-of-domain splits for training and testing. The authors then test 80 models spanning 24 papers on their effectiveness in ranking sequences based on ground truth efficiency scores. Experimental results reveal that BLMs perform well in in-domain ranking, However, BLMs show limited performance for out-of-domain ranking, underscoring a significant challenge in generalizing to unfamiliar data.

**Strengths:**

The paper introduces a dataset that addresses a crucial evaluation criterion (specific functional predictions) for model performance.

The paper evaluates over 80 models across 24 papers spanning protein, DNA, RNA, and multimodal domains which is an impressive and commendable effort, adding significant value to the study.

The paper includes a detailed comparison and analysis of BLMs, examining factors such as modality, model size, and K-mer.

**Weaknesses:**

Questions to improve clarity:

I think the the papers is a bit a hard to read and there are several open questions which could improve the clarity of the paper noted below -

(1) How is the read score representative of the evolution effectiveness? Is the least ranking sequence in the dataset completely ineffective at editing or all the 100K variants effective? Specifically I would like to know if you are also evaluating BLMs capability in predicting both effectiveness and ineffectiveness?
(2) What external evaluation can you provide to show your effectiveness scores are sound and consistent with previous work / distaste ?
(3) Though Figure 4 is a helpful visualization, I think some kind of end to end pipeline of data construction and usage is helpful, currently the main framework is mixed up with the details provided.
(4) Figure 5 shows range of metric value is between (0.82-0.86) for out-of-domain evaluation, the equivalent score range for in-domain evaluation is (0.85-0.92)? Do  you consider this a significant drop?  What is the drop per model, perhaps a scatter plot with x-axis as in-domain performance and y-axis as out of domain performance per model is helpful? Could other factors like  batch-effects etc be effecting the performance of out-of-domain generalization ?

Significance:

Its unclear on how well evaluating on this task will generalize to other functional prediction tasks and what it the overall recommendation of the dataset for future evaluation.

(1) Given that BLMs area able to achieve a given performance in predicting TadA evolution what other related function prediction tasks will this performance generalize to? And other tasks it wont?  What are the limtations of the datasets and What is the recommendation on how the dataset can be used by model developers?

**Questions:**

It would be helpful to receive a response from the authors on the questions described above.

---

> ### Author Response · Authors · 2024-11-25
>
> Thank you for your thoughtful feedback and for recognizing the strengths of our work. We are grateful for your recognition of the **dataset** (addresses a crucial evaluation criterion), **extensive study** (impressive and commendable effort), and **detailed comparison and analysis of BLMs** (examining factors such as modality, model size, and K-mer). We truly appreciate your insights.
>
> Below, please find our detailed responses to your concerns.
>
> ## 1 Bio Experiment
>
> >How is the read score representative of the evolution effectiveness? Is the least ranking sequence in the dataset completely ineffective at editing or all the 100K variants effective? Specifically I would like to know if you are also evaluating BLMs capability in predicting both effectiveness and ineffectiveness?
>
> Thank you for your insightful question. As shown in Appendix A.1, the editing efficiency of TadA protein is directly determined by its enzymatic activity, which in turn influences the replication of bacteriophages. This leads to higher read counts for sequences with higher editing efficiencies. Rather than being a binary classification of “effective” versus “ineffective,” this is a continuous spectrum of activity levels.
>
> In the dataset derived from NGS sequencing, the sequences ranked lowest in terms of read count generally have lower editing efficiency than those ranked higher. However, whether these sequences are entirely inactive is not always clear. This is precisely why we chose to construct the dataset using rankings instead of absolute activity values: rankings focus on the relative performance of sequences, making the data representation more robust against noise and experimental variability.
>
> Thus, the dataset does not evaluate effectiveness as a binary trait but rather as a continuum of activity levels. This approach ensures that the benchmarking remains focused on distinguishing relative performance, which aligns better with the inherent variability of biological experiments.
>
> >What external evaluation can you provide to show your effectiveness scores are sound and consistent with previous work / distaste ?
>
> Thank you for raising this important question. We agree that biological experiments inherently exhibit variability and system-specific fluctuations. In such cases, absolute calibration across different experimental setups is difficult to achieve. Thus, comparisons are typically made relative to well-established baselines or control groups.
>
> In our study, we ensured the reliability of our evolutionary experiments through extensive preliminary trials to validate the experimental setup. Moreover, our experiments began with TadA8e—a globally recognized benchmark enzyme with high activity. This starting point serves as a robust and reliable control, reinforcing the soundness of our methodology. As a result, while absolute calibration may not be feasible, our scores are meaningful and consistent within the context of our controlled experimental framework.
>
> We believe this approach demonstrates the validity and reliability of our findings, aligning with best practices in the field. Thank you for allowing us to clarify this point.
>
> ## 2 SEQ2RANK
> > Though Figure 4 is a helpful visualization, I think some kind of end to end pipeline of data construction and usage is helpful, currently the main framework is mixed up with the details provided.
>
> Thank you for pointing this out. We appreciate your suggestion to clarify the framework further with an end-to-end pipeline illustration. While Figure 4 provides a detailed visualization, we acknowledge that separating the framework from the finer details can enhance the clarity and utility of our presentation.
>
> It is important to note that our approach, SEQ2RANK, inherently serves as an end-to-end pipeline for ranking tasks based on biological sequencing data. From raw sequence input to ranked output, SEQ2RANK integrates data preprocessing, ranking extraction, and consistency maintaining within a cohesive framework. This design emphasizes its generalizability and adaptability to various biological sequencing-rank scenarios, ensuring a seamless workflow for the user.
>
> Further discussion continues on the next page.

---

> > ### Author Response · Authors · 2024-11-25
> >
> > ## 3 ID and OOD
> > >Figure 5 shows range of metric value is between (0.82-0.86) for out-of-domain evaluation, the equivalent score range for in-domain evaluation is (0.85-0.92)? Do you consider this a significant drop? What is the drop per model, perhaps a scatter plot with x-axis as in-domain performance and y-axis as out of domain performance per model is helpful? Could other factors like batch-effects etc be effecting the performance of out-of-domain generalization ?
> >
> > You are correct that the absolute values of the metrics appear close. However, NDCG is a non-linear metric, and for @100, even random initialization achieves a score around 0.84. Thus, improvements beyond this baseline, such as the 0.85–0.92 range in in-domain evaluation, represent meaningful gains. Due to space constraints, we provide detailed data in the appendix, where you can find a thorough breakdown. Additionally, we will include a scatter plot comparing in-domain and out-of-domain (OOD) performance per model in the appendix for better visualization.
> >
> > A more intuitive metric for comparison is the Spearman correlation coefficient (SP). Please refer to the appendix, specifically Table 4 and Table 9, which clearly illustrate the challenges of OOD generalization.
> >
> > Regarding OOD performance, we believe the challenges stem primarily from significant differences in the sequence space distributions between domains. For our experiments, the experimental setup remains fixed, and as we focus on relative rankings, the final high/low relative activity is preserved if sufficient competition is ensured during each experiment. Batch effects may influence datasets with absolute value comparisons, but their impact is less likely in our ranking-focused dataset.
> >
> > ## 4 Application and Generalization
> > >Given that BLMs area able to achieve a given performance in predicting TadA evolution what other related function prediction tasks will this performance generalize to? And other tasks it wont? What are the limtations of the datasets and What is the recommendation on how the dataset can be used by model developers?
> >
> > Thank you for your insightful question. The benchmarked models’ performance in predicting TadA evolution can potentially generalize to related tasks in protein engineering, particularly those involving enzymes or other components of gene-editing systems such as Cas proteins. These tasks share common principles in sequence-function relationships, enabling models trained on TadA data to provide valuable insights. However, generalization to tasks outside RNA or gene editing, or those involving proteins with drastically different structures or mechanisms, may be more limited.
> >
> > Regarding dataset limitations, it is important to note that our data focuses primarily on TadA. The dataset is highly targeted, which limits its immediate applicability to unrelated protein functions. Nevertheless, for model developers, this dataset serves as a robust testbed for evaluating sequence-function prediction models in protein engineering. We recommend using it as a foundation for tasks involving enzymes or extending it by integrating complementary data, such as datasets for Cas proteins or other gene-editing systems, to broaden its scope and enhance model generalization.

---

### Official Review · Reviewer_LQYf · 2024-11-01

**Soundness:** 1
**Presentation:** 2
**Contribution:** 3
**Rating:** 3
**Confidence:** 4

**Summary:**

This paper introduces a new dataset called NOVOBENCH-100K, which contains 101,687 unique DNA variants with an average of 11.1 amino acid mutations. This dataset can be used to evaluate biological language models (BLMs), including pLM, DNALM, and RNALM. The authors assessed 80 BLMs across both in-domain and out-of-domain scenarios. They have released some code and data.

**Strengths:**

(1) Provides a large-scale dataset that can be used for computer evaluation of various machine learning algorithms;
(2) Conducts an extensive study of 80 BLM models in both in-domain and out-of-domain ranking scenarios;
(3) The paper is well written;

**Weaknesses:**

There are several concerns regarding the evaluation presented in this paper:

**Dataset Division**: To rigorously evaluate a machine learning algorithm, it is necessary to prepare separate training, validation, and test sets. However, this work only partitions the dataset into training and test sets. It is not a good practice to use the test set to tune hyperparameters (e.g., batch size, early stopping, or training epochs). The test set here is actually a validation set. A good performance on the validation set does not mean that the results on the test set will be similar.

**Evaluation Method**: The paper primarily evaluates linear probing in the main text, but linear probing typically performs significantly worse than fine-tuning the entire model or the top several layers of the language model. I noticed in the appendix Table 6 that the authors only evaluated ESM2 using both linear probing and all-parameter fine-tuning, revealing a substantial performance gap (e.g., the SP metric increased from 0.138 to 0.208 and from 0.075 to 0.187). Given this difference of pLM, I suggest the authors present the main results for these pLM algorithms using parameter fine-tuning rather than a frozen embedding strategy.

**Evaluation of Multiple Models**: The authors claim to have evaluated up to 80 biological language models (BLMs), which seems quite surprised to me. Each model's hyperparameters should be meticulously tuned. How can the authors ensure that they are using each model correctly or with optimal settings? If suboptimal settings were used, the benchmark may not be reliable. For instance, SaProt is a protein language model that incorporates structural information, yet the paper does not specify what type of structures were used (PDB or AlphaFold2 structures with or without pLDDT). This information is not available in the supplementary materials.Evaluating more models is a good thing, but one should make sure that they actually understand and evaluate them correctly.

**Learning to Rank Method**: The paper introduces a listwise learning-to-rank method for all models and claims this as a contribution. However, have the authors evaluated this ranking loss against regression loss? In fact, findings from the 2011 Yahoo Learning to Rank Challenge indicate that learning to rank may not significantly outperform pointwise regression or classification losses. In some cases, regression loss could be more effective if specific label values are available. see [1] section 6.3. I do not think listNet is a powerful learning to rank algorithm. The author should report convincing results that  listNet is better than basic regression loss.

[1]https://proceedings.mlr.press/v14/chapelle11a/chapelle11a.pdf

**Questions:**

no

---

> ### Author Response · Authors · 2024-11-25
>
> Thank you for your thoughtful feedback and for recognizing the strengths of our work. We are grateful for your recognition of the **large-scale dataset**, **extensive study**, and **well-written paper**. We truly appreciate your insights.
>
> Below, please find our detailed responses to your concerns.
>
> ## 1 Dataset Division
>
> >To rigorously evaluate a machine learning algorithm, it is necessary to prepare separate training, validation, and test sets. However, this work only partitions the dataset into training and test sets. It is not a good practice to use the test set to tune hyperparameters (e.g., batch size, early stopping, or training epochs). The test set here is actually a validation set. A good performance on the validation set does not mean that the results on the test set will be similar.
>
> Thank you for your valuable comment. You are absolutely right—we used the test set for hyperparameter tuning in this work. Moving forward, as we collect more data across additional domains and rounds, we will ensure to properly separate validation and test sets to avoid this issue.
>
> From another perspective, even though we performed hyperparameter tuning on the test set, the poor performance on out-of-domain (OOD) data highlights the severity of the OOD challenge, which remains a key limitation of the current models.
>
> ## 2 Evaluation Method
>
> >The paper primarily evaluates linear probing in the main text, but linear probing typically performs significantly worse than fine-tuning the entire model or the top several layers of the language model. I noticed in the appendix Table 6 that the authors only evaluated ESM2 using both linear probing and all-parameter fine-tuning, revealing a substantial performance gap (e.g., the SP metric increased from 0.138 to 0.208 and from 0.075 to 0.187). Given this difference of pLM, I suggest the authors present the main results for these pLM algorithms using parameter fine-tuning rather than a frozen embedding strategy.
>
> Thank you for your insightful comment. We agree that fine-tuning generally yields better in-domain performance compared to linear probing. However, fine-tuning 80 models is computationally expensive, and we were not able to afford such a comprehensive evaluation within the scope of this work.
>
> Also, our primary focus in this work is to compare the representational power of different large-scale models, and thus we chose to evaluate them using linear probing. This approach allowed us to focus on understanding the quality of embeddings across different models.
> Furthermore, in our experiments, fine-tuning did not significantly improve out-of-domain performance, which remains a key challenge. Therefore, we believe the conclusions drawn from our linear probing experiments still hold strong in terms of supporting our central argument.
>
> ## 3 Evaluation of Multiple Models
>
> >The authors claim to have evaluated up to 80 biological language models (BLMs), which seems quite surprised to me. Each model's hyperparameters should be meticulously tuned. How can the authors ensure that they are using each model correctly or with optimal settings? If suboptimal settings were used, the benchmark may not be reliable. For instance, SaProt is a protein language model that incorporates structural information, yet the paper does not specify what type of structures were used (PDB or AlphaFold2 structures with or without pLDDT). This information is not available in the supplementary materials.Evaluating more models is a good thing, but one should make sure that they actually understand and evaluate them correctly.
>
> Thank you for your thoughtful feedback and recognition of the scope of our work. The core objective of our experiments was to evaluate the representational capabilities of different BLMs. As part of this, we primarily focused on extracting embeddings and then performing linear probing. To ensure we were using each model appropriately, we performed extensive hyperparameter tuning for each model. For linear probing, we carefully tested different learning rates, optimizers, and other hyperparameters, ultimately selecting the best-performing configurations for each model.
>
> In particular, for SaProt, we followed the setting from the original paper’s Appendix E.3, which describes the “RESIDUE SEQUENCE-ONLY” setting. The original paper describes the setting as:
> >...all Foldseek structure tokens are substituted with “#”, resulting in “si#”. This scenario often arises in various protein engineering tasks where the fitness values of protein variants can be obtained through wet experiments, but experimental structures for them are unavailable.
>
> Further discussion continues on the next page.

---

> ### Author Response · Authors · 2024-11-25
>
> ## 4 Learning to Rank
>
> >The paper introduces a listwise learning-to-rank method for all models and claims this as a contribution. However, have the authors evaluated this ranking loss against regression loss? In fact, findings from the 2011 Yahoo Learning to Rank Challenge indicate that learning to rank may not significantly outperform pointwise regression or classification losses. In some cases, regression loss could be more effective if specific label values are available. see [1] section 6.3. I do not think listNet is a powerful learning to rank algorithm. The author should report convincing results that listNet is better than basic regression loss.
> [1]https://proceedings.mlr.press/v14/chapelle11a/chapelle11a.pdf
>
> Thank you for your valuable feedback and for pointing out the work from the 2011 Yahoo Learning to Rank Challenge. We agree that learning-to-rank (LTR) methods may not always outperform regression losses, especially when specific label values are available.
>
> However, our motivation for using a listwise LTR approach was based on the fact that our dataset provides *only relative ranking information (i.e., the order within a list)*, rather than explicit numeric labels. It is because relative ranking rather than absolute numeric values, helps mitigate random noise, experimental variability, and discrepancies across different rounds of evolution. Therefore, traditional regression approaches would not be suitable in this case, as they require exact target values for training.
>
> In Table 2 of the Appendix, we explored several loss functions related to ranking, including Kullback-Leibler (KL) divergence and cosine similarity. Ultimately, we selected cross-entropy loss for our ranking tasks, as it provided a good balance between performance and theoretical grounding for listwise ranking.

---

> ### Comment · Reviewer_LQYf · 2024-11-26
> **Thanks**
>
> Thank you for your response. However, there still remain some concerns that need to be addressed:
>
> 1.	Regarding Hyperparameter searching on the test set.
>
> The current approach of using the test set for hyperparameter optimization raises significant concerns and potentially compromises the validity of the results. Your proposed solution of collecting additional data in the future does not address this issue with the current benchmark. As a benchmark study, ensuring the reliability and reproducibility of your evaluation framework is paramount.
>
> 2.	Limited Evaluation Scope.
>
> Your study evaluates 80 models but primarily under linear probing conditions. While this broad comparison is interesting, the absence of fine-tuning results - which typically yield better performance in practice - limits your conclusions. I suggest focusing on fewer models but evaluating them under optimal fine-tuning settings, as this would provide more actionable insights than testing many models under suboptimal conditions.
>
> 3.	Baseline Issues.
>
> Upon reviewing your evaluation details, let's say SaProt, there appears to be a discrepancy. The original paper specifies that SaProt works on residue-only sequences during the fine-tuning stage. In your implementation, the SaProt model remains frozen. As a structure-aware PLM claimed by the paper, simply removing its structure part without any fine-tuning may not make sense.
>   It would be beneficial to:
> • Consider implementing the fine-tuning stage as described in the paper
> • Provide some necessary analysis of why some baselines perform differently under your specific conditions.
>
> Comparing baselines under fair settings is important for a benchmark paper.

---

### Official Review · Reviewer_jE3h · 2024-11-02

**Soundness:** 4
**Presentation:** 3
**Contribution:** 4
**Rating:** 6
**Confidence:** 3

**Summary:**

This paper introduces NOVOBENCH-100K, a large-scale dataset aimed at advancing the in silico evolution of the TadA enzyme, a critical component in base editing. This dataset is unique as it provides 101,687 DNA variants and employs a novel ranking-based approach, SEQ2RANK, rather than traditional classification labels, to account for the biological experiment's credibility and ranking consistency. The authors also benchmark 80 biological language models (BLMs) across protein, DNA, RNA, and multimodal domains, demonstrating these models' proficiency in in-domain tasks while highlighting challenges in out-of-domain ranking.

**Strengths:**

Novel Dataset: The creation of NOVOBENCH-100K fills a crucial gap in protein engineering, specifically for base editing.
Robust Methodology: The SEQ2RANK algorithm ensures data consistency and robustness, addressing typical noise and variability in biological data.
Comprehensive Benchmarking: The inclusion of various BLMs and modalities provides a detailed performance landscape, especially in distinguishing in-domain from out-of-domain capabilities.

**Weaknesses:**

Limited Real-World Application Validation: Although the dataset is extensive, there is limited discussion on how it performs in real-world applications beyond the in silico setting. Including preliminary results from practical applications or collaborations with experimental labs could strengthen the paper.
Out-of-Domain Generalization: The authors highlight the poor performance of current BLMs on out-of-domain rankings. However, the paper could benefit from a more in-depth analysis of why these models fail and potential strategies for overcoming these limitations.
Benchmarking Scope: Although 80 models are benchmarked, it would be beneficial to include additional model families or discuss other state-of-the-art models in more detail to contextualize their choices further.

**Questions:**

Can the authors clarify if SEQ2RANK can be generalized to other protein datasets or if it’s specifically tailored for the TadA evolution? This would help in understanding its broader applicability.
Are there plans to address the out-of-domain challenges identified in the paper, perhaps by incorporating domain adaptation techniques or more diverse training data?

---

> ### Author Response · Authors · 2024-11-25
>
> Thank you for your thoughtful feedback and for recognizing the strengths of our work. We are grateful for your recognition of the **Novel Dataset**, **Robust Methodology**, and **Comprehensive Benchmarking**. We truly appreciate your insights.
> >The creation of NOVOBENCH-100K fills a crucial gap in protein engineering, specifically for base editing.
>
> >The SEQ2RANK algorithm ensures data consistency and robustness, addressing typical noise and variability in biological data.
>
> >The inclusion of various BLMs and modalities provides a detailed performance landscape, especially in distinguishing in-domain from out-of-domain capabilities.
>
> Below, please find our detailed responses to your concerns.
>
> ## 1 Application
>
> >Limited Real-World Application Validation: Although the dataset is extensive, there is limited discussion on how it performs in real-world applications beyond the in silico setting. Including preliminary results from practical applications or collaborations with experimental labs could strengthen the paper.
>
> Thank you for your suggestion. The real-world applications of this dataset involve protein engineering and directed evolution. Researchers can use our dataset to develop protein evolution algorithms that better mimic real-world multi-round evolution processes.
>
> In fact, using this dataset and the associated algorithms, we have successfully engineered proteins with higher activity than TadA8e. However, due to the scope of this work, we did not include these results in the NovoBench-100K paper. *Additionally, we have no intention of using such results to enhance the recognition of NovoBench-100K, which is focused on the dataset and benchmark itself.*
> Further application-focused outcomes will be released in subsequent work as part of this series. For those interested, we encourage following our future publications.
>
> Our goal in this paper is to provide an open and versatile resource, enabling researchers to freely explore AI-guided protein engineering and related applications. We believe this approach will maximize the potential impact of our work in advancing real-world applications.
>
> ## 2 OOD
> >Out-of-Domain Generalization: The authors highlight the poor performance of current BLMs on out-of-domain rankings. However, the paper could benefit from a more in-depth analysis of why these models fail and potential strategies for overcoming these limitations.
>
> >Are there plans to address the out-of-domain challenges identified in the paper, perhaps by incorporating domain adaptation techniques or more diverse training data?
>
> Thank you for pointing this out. We believe the vastness of the protein sequence space is a key factor contributing to the poor out-of-domain performance of current BLMs. When sequences from different rounds of evolution exhibit significant differences, it becomes challenging for these models to generalize effectively.
>
> To address this, future strategies might include developing methods for constructing latent spaces or adaptive approaches that are inherently more robust to sequence variation. Such advancements could help reduce the impact of sequence discrepancies and improve generalization across rounds. We appreciate this suggestion and will explore these directions in our future work.
>
> Additionally, we are continuously expanding our dataset. Currently, we have collected data from five rounds of evolution experiments, comprising over 300K sequences. We plan to update and release new data regularly to support further OOD research and improve model robustness.
>
> ## 3 More Models
> >Benchmarking Scope: Although 80 models are benchmarked, it would be beneficial to include additional model families or discuss other state-of-the-art models in more detail to contextualize their choices further.
>
> Thank you for your feedback. We have benchmarked 80 models, including cutting-edge and influential models such as ESM3, EVO, Chai1, and OmniGenome. We believe this represents a comprehensive scope of current state-of-the-art models.
> We are more than willing to test additional models and greatly appreciate any specific papers or model families you could point us to. This would further enhance the diversity and impact of our benchmark.
>
>
> ## 4 SEQ2RANK Generation
> >Can the authors clarify if SEQ2RANK can be generalized to other protein datasets or if it’s specifically tailored for the TadA evolution?
>
> Thank you for your question. SEQ2RANK is not specifically tailored to TadA evolution. It is a general framework designed to bridge the gap between raw biological sequencing data (e.g., from NGS) and ranked datasets. As such, it can be applied to any protein dataset where ranking based on activity or other metrics is required.
>
> The method is protein-agnostic and aims to address the broader challenge of converting sequencing data into robust and usable rank-based datasets, enabling its use across various protein engineering and evolution applications.

---

### Official Review · Reviewer_93j9 · 2024-11-04

**Soundness:** 3
**Presentation:** 3
**Contribution:** 3
**Rating:** 6
**Confidence:** 4

**Summary:**

This paper presents a large-scale dataset comprising approximately 100k DNA variants for in silico TadA evaluation, using this dataset to benchmark 80 BLMs. The data, collected from PANCE experiments, is organized into consistent ranking lists of varying lengths (2, 10, and 100) using a novel algorithm, SEQ2RANK. SEQ2RANK mitigates conflicts at the experiment level by prioritizing experiments based on credibility and ensures strict sequence-level consistency through a DAG. This dataset is used to assess model capability via ranking tasks and includes two train-test splits: in-domain and out-of-domain. The BLMs are tested through linear probing and fine-tuning. In the ranking task, where BLMs are used as encoders, model embeddings generally outperform one-hot embeddings. Experiments across modalities show that Chai1’s performance is consistent across different modalities, while LucaOne performs better on proteins than nucleotides. The performance of models of various sizes also demonstrates scaling laws across protein, DNA, and RNA modalities. However, the BLMs perform poorly on the out-of-domain split, highlighting limitations in their practical robustness.

**Strengths:**

1.	The paper conducts extensive experiments to evaluate the capabilities of BLMs on TadA. The authors tested a diverse set of 80 BLMs spanning 24 papers on the NOVOBENCH-100K dataset. These BLMs represent a range of modalities, including: protein, DNA, RNA, multimodal BLMs. The authors evaluated these BLMs using both linear probing and fine-tuning. Authors also tested three different learning rates (1e-5, 1e-4, and 1e-3) for each experiment and reported the best result. The evaluation across a wide range of BLMs, evaluation methods, and hyperparameters contributes to a comprehensive evaluation of BLMs on the TadA protein evolution task.
2.	The NOVOBENCH-100K dataset originates from two rounds of standardized PANCE experiments and includes 100k sequences, making it valuable for research in this domain. The authors also promise to include more rounds of wet experiment data.
3.	The two rounds of experiments provide a useful approach for creating an out-of-domain train-test split. This split offer a practical perspective on model’s robustness in the real-world applications.
4.	The dataset organizes sequences into ordered lists of varying lengths, rather than using scores or labels, to mitigate experimental noise and address scalability limitations. The authors propose SEQ2RANK, which incorporates the credibility of each experiment to ensure the consistency of the ranking data.
5.	The paper is clearly written and well structured. The presentation is pretty good.

**Weaknesses:**

1.	The experiments primarily use BLMs as encoders, leveraging the embeddings for downstream ranking tasks. However, there are other common approaches, such as prompting and in-context learning, which are now widely used. The authors should consider including these methods in the evaluation.
2.	There are additional aspects that could be included in the analysis, such as details about the model architectures (e.g., encoder-decoder or decoder-only). Additionally, providing more description of the initial design purpose of each model and how this aligns with the TadA evaluation would be helpful.
3.	It would be beneficial to conduct more rounds of experiments, as I am questioning whether the model's poor performance on out-of-domain cases is due to overfitting to a specific experimental round. I notice that the author plans to include additional rounds of wet experiment data, which I am looking forward to seeing.
4.	The authors should include detailed descriptions of the wet experiment conditions as a reference for researchers who wish to replicate or expand this dataset.

**Questions:**

1.	There is an existing benchmark names NOVOBENCH (without the 100k) (https://github.com/Westlake-OmicsAI/NovoBench). What is the relationship between NOVOBENCH and NOVOBENCH-100K?
2.	What are your insights on why all models fail in out-of-domain evaluation, and how can the models' robustness be improved for out-of-domain cases?
3.	Why do conflicts arise in the rankings? If these conflicts are due to experimental errors, should the algorithm identify and exclude erroneous experiments from the data, rather than simply using credibility to mitigate the conflicts?

---

> ### Author Response · Authors · 2024-11-25
>
> Thank you for your thoughtful feedback and for recognizing the strengths of our work. We are grateful for your recognition of the
> - **extensive experiments** (a wide range of BLMs, evaluation methods, and hyperparameters),
> - **ranks and SEQ2RANK** (mitigate experimental noise and address scalability limitation),
> - **valuable dataset** (valuable for research in this domain, OOD practical perspective on model’s robustness in the real-world applications),
> - and **paper writing** (presentation is pretty good). We truly appreciate your insights.
>
> Below, please find our detailed responses to your concerns.
>
> ## 1 Evaluation
>
> >The experiments primarily use BLMs as encoders, leveraging the embeddings for downstream ranking tasks. However, there are other common approaches, such as prompting and in-context learning, which are now widely used. The authors should consider including these methods in the evaluation.
>
> Thank you for highlighting this important point. Our primary goal in this work was to broadly evaluate the representation capabilities of biological language models (BLMs). To this end, we focused on approaches like linear probing and fine-tuning, prioritizing experiments across a wider range of models.
>
> We agree that prompting and in-context learning are valuable methods and offer exciting possibilities for leveraging BLMs. While they were beyond the scope of this work, we plan to explore these approaches in future studies. Thank you again for the suggestion!
>
> >There are additional aspects that could be included in the analysis, such as details about the model architectures (e.g., encoder-decoder or decoder-only). Additionally, providing more description of the initial design purpose of each model and how this aligns with the TadA evaluation would be helpful.
>
> Thank you for emphasizing the importance of providing detailed model architecture and background information. We recognize that these aspects are critical for a comprehensive understanding of the models used.
> We have already compiled substantial details about the models, including architecture, training datasets, and design purposes. However, due to space constraints, we were unable to include them in the main text. To ensure transparency and provide these details to the community, we plan to make them publicly available on an open-source platform (e.g., GitHub).
>
> Below is a sample of the compiled information we plan to release in markdown format:
>
> | Modality | Model Name                | Training Data Size                  | Data Source Description                                | Model Size |
> |----------|---------------------------|-------------------------------------|------------------------------------------------------|------------|
> | Protein  | esm2_t6_8M_UR50D          | Billions of protein sequences       | Various eukaryotic, prokaryotic, and viral proteins  | 8M         |
> | Protein  | esm3_sm_open_v1           | 771 billion tokens                 | Natural proteins, excluding toxins and viruses       | 1.4B       |
> | Protein  | prot_bert                 | 217 million sequences              | UniProt, a comprehensive protein database            | 420M       |
> | Protein  | ProstT5                   | 17M proteins                       | 3Di+AA sequences for 17M proteins                   | 3B         |
> | Protein  | prot_t5_xl_uniref50       | 45M sequences                      | UniRef50, from various organisms                    | 3B         |
> | DNA      | EVO131k                   | 300B tokens                        | Whole prokaryotic genomes                           | 6.45B      |
> | DNA      | LucaOne_DNA               | -                                  | 169,861 species (proteins, DNA, RNA)                | 1.8B       |
> | DNA      | 500M_multi_species_v2     | 174B nucleotides, 29B tokens       | NCBI data across genomes (human + other species)    | 498M       |
> | RNA      | mrnafm                    | 23.7M RNA sequences                | RNAcentral                                          | 241M       |
> | RNA      | rnabert                   | 722,370 RNA sequences              | Human ncRNA                                         | 533k       |
> | RNA      | calm                      | 9.8M cDNA sequences -> RNA data    | ENA, encoding sequences from various organisms      | 85.7M      |
>
> Further discussion continues on the next page.

---

> ### Author Response · Authors · 2024-11-25
>
> ## 2 More Experiments and Details
>
> >It would be beneficial to conduct more rounds of experiments, as I am questioning whether the model's poor performance on out-of-domain cases is due to overfitting to a specific experimental round. I notice that the author plans to include additional rounds of wet experiment data, which I am looking forward to seeing.
>
> Thank you for your thoughtful feedback. As of now, we have conducted 5 rounds of evolution experiments, resulting in 348,267 sequences and 344,497,666 edges in a directed graph. This extensive dataset allows us to evaluate the model’s performance across diverse scenarios, including out-of-domain cases.
> We are actively exploring additional ML experiments while collecting more data, and further results will be shared in our future work. We appreciate your interest and will ensure transparency in reporting updates as they become available.
>
> >The authors should include detailed descriptions of the wet experiment conditions as a reference for researchers who wish to replicate or expand this dataset.
>
> Thank you for your suggestion. We have provided the wet experiment methods in Appendix A.1. However, the specific experimental parameters are currently proprietary to our collaborating company and are not being made publicly available at this stage.
>
> We are committed to transparency and reproducibility, and as such, we will release the complete dataset, AI code, and BLM embeddings to support further research and replication efforts. Should the company decide to disclose these details in the future, we will ensure they are made available.
>
> ## 3 Benchmark Name
> >There is an existing benchmark names NOVOBENCH (without the 100k) (https://github.com/Westlake-OmicsAI/NovoBench). What is the relationship between NOVOBENCH and NOVOBENCH-100K?
>
> Thank you for bringing this to our attention. We acknowledge that the name “NovoBench-100K” coincides with the existing “NovoBench” benchmark. This similarity is unintentional, and we apologize for any confusion it may have caused.
>
> ## 4 Bio Experiments
> >What are your insights on why all models fail in out-of-domain evaluation, and how can the models' robustness be improved for out-of-domain cases?
>
> Thank you for this insightful question. We believe that the vastness of the protein sequence space poses a significant challenge, especially when sequences from different rounds of evolution exhibit notable differences. This divergence is difficult for current BLM approaches to fully address.
>
> In the future, simpler downstream or adaptation methods for constructing latent spaces or adaptive approaches that are more robust to sequence variation may help mitigate the impact of these differences. Such methods could reduce the effect of sequence discrepancies across rounds and improve model performance in out-of-domain scenarios.
>
> >Why do conflicts arise in the rankings? If these conflicts are due to experimental errors, should the algorithm identify and exclude erroneous experiments from the data, rather than simply using credibility to mitigate the conflicts?
>
> Thank you for this insightful question. Ranking conflicts can arise due to inherent variability and uncertainty in biological experiments, which are influenced by factors such as noise, experimental conditions, and measurement errors. Rather than excluding data outright, which risks losing valuable information, we opted to design a framework that assigns credibility scores to account for these conflicts.
>
> To clarify, we conducted two separate rounds of large-scale evolution experiments (R1 and R2), each consisting of multiple turns over time (T1, T2, T3, …). The credibility score leverages the fact that E. coli growth rate is proportional to enzyme activity. Over time (T1, T2, T3, ...), this leads to an exponential increase in read counts for sequences with higher activity, which reduces the impact of initialization differences and random noise. This makes later-round data more reliable.

---

> > ### Comment · Reviewer_93j9 · 2024-12-02
> >
> > Thank you for providing additional information. I will keep my original scores.

---

### Official Review · Reviewer_xLJx · 2024-11-04

**Soundness:** 2
**Presentation:** 2
**Contribution:** 2
**Rating:** 3
**Confidence:** 4

**Summary:**

This paper introduces NOVOBENCH-100k, a very large-scale benchmark dataset for a base editing enzyme TadA derived from in vitro evolution experiments in wet lab. The experimental data covers 100k DNA variants with an average of 11.1 amino acid mutations, exceeding existing datasets for protein function prediction. The authors generated multiple ranking lists of sequences as opposed to reporting absolute read count as labels, and proposed an algorithms SEQ2RANK that creates partial ranking lists of sequences with the aim to improve ranking consistency at experiment-level and sequence-level. They benchmarked 80 biological language models spanning different sequence modalities on both in-distribution and out-of-distribution train-test splits, and showed preliminary results that BLM fails to predict well in out-of-distribution setting.

**Strengths:**

The paper introduced a new benchmark dataset for protein function prediction which excel at its scale and data quality. With 2 rounds of in-vitro evolution of TadA, the authors created a dataset with large amount of mutations and diversity. Such real-world wet-lab experiments data could be a valuable source of data for protein design community. The data is at DNA level and can be used for multiple biological sequence modalities including DNA, RNA and protein sequences. The introduction of ranking list as opposed to absolute value is not new, but the algorithm for list construction does take into consideration several realistic challenges.

The benchmarking effort is also very extensive, covering 80 BLMs and both in-distribution and out-of-distribution. The OOD setting with different evolution round is a novel setup.

**Weaknesses:**

The data could be valuable source for MLCB community, however since it is considering one specific protein TadA and base editing enzymatic activity, the generalizability could be limited. Moreover, the majority of the work focuses on data creation and benchmarking, the technical contribution and novelty may be insufficient for an ML venue. The idea of using ranking instead of absolute value is introduced in many prior work.

The key technical contribution, Seq2Rank algorithm, is introduced in 3.3, but the writing is too high-level and lacks rigorous definition and mathematical justification. Significant amount of details is put into appendix, Algogirhtm1, which is still too abstract and could not justify for the claim of an effective algorithm that address experimental sensitivity.

According to algo1, it sorts list of experiments by reliability and go over the list of lists from high to low order. Within each list, it constructs dictionary of read counts and apply some greedy sampling strategy to sample “strict” partial ordered list that only contain single sequence for a given count, without introducing cyclic loops into the DAG constructed from data that are already been processed. The “GREEDY_SAMPLE” algorithm is too abbreviated and lacks details, such as how are the read count index sampled at each step, and with what “greedy” order it traverse the possible sequence space.

The proposed SEQ2RANK could also introduce biases. For example, strictly enforcing sampling one sequence at each bin breaks the original value distribution of the reads. It is also not clear whether the generated list can cover similar value range, or some of them will be biased towards higher region. When adding sequences from different experiments into same graph, it implicitly tries to compare read counts across different experiments, which may not be ideal.

Additionally, the algorithm requires definition of reliability from biological priors. The later experiment is strictly more important regardless of the read count value, which may not be true all the time. What if such prior does not exist and there is naturally uncertainty in two replicated experiments that needs to be considered equally?

Figure 4 is not informative about Seq2rank and does not have good correspondence with algorithm 1 in appendix.

**Questions:**

1.	When talking about credibility, it mentions “later rounds are more reliable”. Is this suggesting that read counts across different rounds are compared? E.g., it might compare the read count of seq A in round 1 with the read count of seq B in round 2?
2.	What if there are experimental replicas that may have a batch effect but we don’t necessarily have a “reliability” measure of the replicas and want to consider them as equally important?
3.	If seq A has a very high value in round 1, but is somehow close to noise low value in round 2, given that round 2 is always given higher reliability seq A will be ranked low?  What if the round 2 data is noise, and such an assignment might create a lot of wrong ordering of sequences relative to seq A?
4.	How do we determine what read count index to use in each step of GREED_SAMPLE?

---

> ### Author Response · Authors · 2024-11-25
>
> Thank you for your thoughtful feedback and for recognizing the strengths of our work. We are grateful for your recognition of the **dataset** (scale, quality, and real-world wet-lab data), **algorithm** (for realistic challenges), and **benchmarking effort** (80 BLMs, novel OOD setting). We truly appreciate your insights.
> >The paper introduced a new benchmark dataset for protein function prediction which excel at its scale and data quality.
>
> >Such real-world wet-lab experiments data could be a valuable source of data for protein design community.
>
> >The algorithm for list construction does take into consideration several realistic challenges.
>
> >The benchmarking effort is also very extensive, covering 80 BLMs and both in-distribution and out-of-distribution. The OOD setting with different evolution round is a novel setup.
>
> Below, please find our detailed responses to your concerns.
>
> ## 1 Technical Contribution
> >since it is considering one specific protein TadA and base editing enzymatic activity, the generalizability could be limited.
>
> >the majority of the work focuses on data creation and benchmarking, the technical contribution and novelty may be insufficient for an ML venue.
>
> Thank you for pointing this out. We acknowledge that our work focuses on the specific case of TadA and its base-editing enzymatic activity. However, this choice was intentional to demonstrate the potential of integrating ML methodologies with biological data collection and interpretation in a concrete, well-studied context. While generalizability to other proteins is indeed an important direction for future work, our approach aims to establish a foundational framework. For example, the focus on real-world biological scenarios like out-of-distribution (OOD) protein evolution and robust ranking from sequencing data highlights the broader applicability of ML to address key challenges in biology.
>
> We believe such integration is essential for enabling ML for Science, rather than merely applying ML techniques to isolated datasets.
> We emphasize that this effort reflects a deliberate integration of ML principles into the process of biological data generation and curation, rather than treating the two fields in isolation. In our view, separating biological data collection from ML research risks creating a “science + ML” dynamic, rather than fostering true “ML for Science.” Specifically, our contributions concerning dataset construction include:
> - Designing a pipeline that captures realistic OOD protein evolution scenarios.
> - Using ML-inspired strategies to ensure the robustness of ranking lists derived from noisy biological sequencing data.
> - Demonstrating how such datasets can lead to meaningful insights and evaluations for ML models in scientific contexts.
>
>
> ## 2 Seq2Rank Details
>
> Thank you for pointing this out. We will clarify these details in the revised manuscript. Here are short responses to each question:
>
> >The “GREEDY_SAMPLE” algorithm is too abbreviated and lacks details, such as how are the read count index sampled at each step, and with what “greedy” order it traverse the possible sequence space.
>
> >How do we determine what read count index to use in each step of GREED_SAMPLE?
>
> The ‘GREEDY_SAMPLE’ algorithm selects reads based on the highest current read count at each step. We build the "read-seqs dictionary", Dict[float, List[str]]. Then, we choose the read index with the most number of seqs, i.e., GREEDY_SAMPLE (``max(read_seqs_dict.keys(), key=lambda x: len(read_seqs_dict[x]))``). Furthermore, once a sequence is selected to the list, it is removed from its corresponding read-seq dictionary, dynamically updating the read counts. This ensures that the algorithm maximizes the extraction of the most abundant and representative lists.
>
>
> >It is also not clear whether the generated list can cover similar value range, or some of them will be biased towards higher region. When adding sequences from different experiments into same graph, it implicitly tries to compare read counts across different experiments, which may not be ideal.
>
> Thank you for raising this important question. Indeed, using raw values to construct datasets often introduces biases, as value distributions can vary significantly across experiments. To address this, we use ranked lists, ensuring that the relative activity within each list remains accurate. When combining rank lists from different experiments into the graph, we do not rely on absolute values; only relative order is preserved.
>
> By maintaining the correctness of relative rankings within each list and leveraging Seq2Rank to ensure consistency across lists, we effectively mitigate the issue of inconsistencies in raw values from different experiments. While the rank-based approach inherently diverges from the original value distribution, it enables robust comparisons and avoids biases introduced by value range mismatches.
>
> Further discussion continues on the next page.

---

> ### Author Response · Authors · 2024-11-25
>
> >Figure 4 is not informative about Seq2rank and does not have good correspondence with algorithm 1 in appendix.
>
> Thank you for your feedback. We will ensure better clarity by aligning Figure 4 with Algorithm 1 and improving its correspondence in the revised manuscript. Additionally, we will release all code for transparency and include key Python implementation details in the supplementary material to provide further clarity on Seq2Rank.
>
> ## 3 Bio Experiment Details
>
> >What if such prior does not exist and there is naturally uncertainty in two replicated experiments that needs to be considered equally?
>
> >What if there are experimental replicas that may have a batch effect but we don’t necessarily have a “reliability” measure of the replicas and want to consider them as equally important?
>
> Thank you for this thoughtful question. Our experimental design couples enrichment levels with enzymatic activity, ensuring that over successive rounds of evolution, enzymes with different activity levels increase exponentially in number. This design helps mitigate uncertainty by leveraging the underlying biological principles.
>
> We fully acknowledge that biological experiments inherently involve uncertainty and randomness, and no approach can entirely eliminate these challenges. Our method proposes a framework to assess and reduce variability across experiments by emphasizing reliability. However, we also recognize that in cases of perfectly replicated experiments yielding different results, the standard practice of averaging over multiple repetitions remains the most practical solution, albeit imperfect.
>
> >When talking about credibility, it mentions “later rounds are more reliable”. Is this suggesting that read counts across different rounds are compared? E.g., it might compare the read count of seq A in round 1 with the read count of seq B in round 2?
>
> >If seq A has a very high value in round 1, but is somehow close to noise low value in round 2, given that round 2 is always given higher reliability seq A will be ranked low? What if the round 2 data is noise, and such an assignment might create a lot of wrong ordering of sequences relative to seq A?
>
> We apologize for the confusion caused by our wording. To clarify, we conducted two separate rounds of large-scale evolution experiments (R1 and R2), each consisting of multiple turns over time (T1, T2, T3, …).
>
> Within a single evolution experiment, later rounds (e.g., T5) benefit from the coupling of enzymatic activity with the exponential growth of E. coli. This effect amplifies the distinction in read counts between enzymes with differing activity levels, making the data from later rounds more reliable. We will revise the manuscript to ensure this explanation is clearer.

---

### Meta-Review · Area_Chair_MCFZ · 2024-12-17

**Metareview:**

This paper presents a large dataset and a ranking-based evaluation scheme for measuring biological language models’ (BLMs) capability in predicting how variants of TadA perform in editing DNA. They test 80 BLMs and find them okay in-domain but poor out-of-domain.

During the rebuttal, the authors did reply to reviewers and tried to clarify their algorithm (SEQ2RANK) and the dataset details. They also promised to release more data and code. They addressed some points but did not fundamentally change reviewers’ reservations.

The reviewers don't reached a comfortable agreement. Some liked the dataset scale and the broad benchmarking, but others felt that the technical novelty was limited, the evaluation questionable (no proper validation/test split), and the chosen approach (ranking with no separate validation) was not ideal. They remained somewhat non-positive overall.

Personally I feel the future iteration may concern: (a) the dataset splitting and evaluation protocol is questionable—tuning hyperparameters on the test set is not proper;
(b) a need for more rigorous evaluation settings (like proper fine-tuning and carefully chosen baselines) rather than just linear probing and vague architecture documentation. These current issues weaken the reliability and clarity of the benchmark results.

**Additional Comments On Reviewer Discussion:**

see above

---

### Decision · Program_Chairs · 2025-01-22

Reject